# No Change of *Pneumocystis jirovecii* Pneumonia after the COVID-19 Pandemic: Multicenter Time-Series Analyses

**DOI:** 10.3390/jof7110990

**Published:** 2021-11-19

**Authors:** Dayeong Kim, Sun Bean Kim, Soyoung Jeon, Subin Kim, Kyoung Hwa Lee, Hye Sun Lee, Sang Hoon Han

**Affiliations:** 1Department of Internal Medicine, Division of Infectious Disease, Yonsei University College of Medicine, 211 Eonju-ro, Gangnam-gu, Seoul 06273, Korea; DAYOUNG747@yuhs.ac (D.K.); SUBINK93@yuhs.ac (S.K.); KHLEE0309@yuhs.ac (K.H.L.); 2Department of Internal Medicine, Division of Infectious Diseases, Korea University College of Medicine, 73, Goryeodae-ro, Seongbuk-gu, Seoul 02841, Korea; puppybin@gmail.com; 3Biostatistics Collaboration Unit, Yonsei University College of Medicine, 211 Eonju-ro, Gangnam-gu, Seoul 06273, Korea; JSY0331@yuhs.ac

**Keywords:** COVID-19, non-pharmacological interventions, *Pneumocystis jirovecii*, pandemic, time-series analysis

## Abstract

Consolidated infection control measures imposed by the government and hospitals during COVID-19 pandemic resulted in a sharp decline of respiratory viruses. Based on the issue of whether *Pneumocystis jirovecii* could be transmitted by airborne and acquired from the environment, we assessed changes in *P. jirovecii* pneumonia (PCP) cases in a hospital setting before and after COVID-19. We retrospectively collected data of PCP-confirmed inpatients aged ≥18 years (*N* = 2922) in four university-affiliated hospitals between January 2015 and June 2021. The index and intervention dates were defined as the first time of *P. jirovecii* diagnosis and January 2020, respectively. We predicted PCP cases for post-COVID-19 and obtained the difference (residuals) between forecasted and observed cases using the autoregressive integrated moving average (ARIMA) and the Bayesian structural time-series (BSTS) models. Overall, the average of observed PCP cases per month in each year were 36.1 and 47.3 for pre- and post-COVID-19, respectively. The estimate for residuals in the ARIMA model was not significantly different in the total PCP-confirmed inpatients (7.4%, *p* = 0.765). The forecasted PCP cases by the BSTS model were not significantly different from the observed cases in the post-COVID-19 (−0.6%, 95% credible interval; −9.6~9.1%, *p* = 0.450). The unprecedented strict non-pharmacological interventions did not affect PCP cases.

## 1. Introduction

Stringent policies for social or physical distancing, universal facial masking, travel restriction, self-isolation, and quarantine are globally being implemented as important and effective strategies to mitigate the COVID-19 pandemic in community settings [1,2]. Additionally, the operation of triage centers and respiratory safety clinics, restriction of visitors, and anticipative tests of severe acute respiratory syndrome coronavirus 2 (SARS-CoV-2) in health care workers (HCWs) in close contact with COVID-19 patients or routine monitoring of asymptomatic high-risk HCWs have been the main protective measures to prevent intra-hospital outbreaks in crowded hospital settings [3,4,5]. This strategy towards “zero COVID” eventually led to a decrease in social outdoor activities and an increase in time spent at home [6,7].

Non-pharmacological interventions (NPIs), including respiratory and hand hygiene in both public and hospitals, can reduce the incidence of community-acquired respiratory viruses (CA-RVs) with seasonal influenza that cause airborne person-to-person transmission through respiratory droplets and/or aerosols to varying degrees [8,9,10,11]. The unprecedented efforts of NPIs during the COVID-19 pandemic resulted in a marked decrease in influenza incidence in the general population from as early as January 2020 during the early COVID-19 outbreak and which endured until the latter end of the northern or southern hemisphere’s 2020–2021 influenza seasons [12,13,14,15,16,17]. Recently, a few studies reported that the detection of various CV-RVs (adenovirus, bocavirus, enterovirus, influenza A/B, metapneumovirus, non-SARS-CoV-2 coronavirus, parainfluenza virus, rhinovirus, and respiratory syncytial virus (RSV)) was significantly and continuously decreased within a short time after national social distancing or lockdown in UK inpatients with hematologic diseases and in children in Australian communities [18,19].

*Pneumocystis jirovecii* is an important respiratory opportunistic pathogen that contributes to critical illness and poor outcomes in severe immunocompromised transplant recipients and patients with acquired immunodeficiency syndrome [20]. Even though *P. jirovecii* pneumonia (PCP) may occur after environmental acquisition [20,21,22,23,24,25,26], accumulating evidence has established person-to-person transmission (possibly airborne) or asymptomatic carriers (colonization) of *P. jirovecii*, even in the general population, strongly supported by many outbreaks in healthcare-associated and community environments [27,28,29,30,31,32,33,34,35,36,37]. Several seroepidemiologic surveys revealed that the majority (70–80%) of healthy children were seropositive for *P. jirovecii* [38,39,40]. *P. jirovecii* was detected in air samples within 1 to 8 m from patients with PCP or *P. jirovecii* colonization and intensive care unit (ICU) HCWs, suggesting airborne spread [32,34,36,41,42,43]. Additionally, recent analyses with metagenomic data sets have focused on the frequent air shedding of *P. jirovecii* and their transmission from PCP-confirmed patients, supporting early epidemiologic data [44].

Based on this recent substantial proof and these distinctive features of *P. jirovecii* as a transmissible fungus, we hypothesized that the multifarious infection control measures of enforced-NPIs with strong compliance during the COVID-19 pandemic could prevent the intra-hospital transmission and community acquisition of *P. jirovecii*. Therefore, we investigated the change in PCP cases in the COVID-19 pandemic era to evaluate the effect of strict NPIs, except contact isolation, on PCP incidence.

## 2. Materials and Methods

### 2.1. Study Design and Data Collection

We performed a multicenter retrospective longitudinal observational study in four university-affiliated tertiary general hospitals: Yonsei University Severance Hospital, Gangnam Severance Hospital, and Yongin Severance Hospital (2500, 800, and 500 beds, respectively) and Korea University Anam Hospital (1100 beds), in Seoul and Gyeonggi-do. We detected *P. jirovecii* with polymerase chain reaction (PCR) and cytology tests in the sputum, bronchial washing, and bronchoalveolar lavage fluid specimens and histopathology in lung and bronchus tissues from patients aged ≥18 years between 1 January 2015 and 30 June 2021, using query-based relational database management systems (RDBMS). We additionally sorted the cytology and histopathology results with Giemsa or Grocott-Gomori methenamine silver staining through a text search in the RDBMS. The diagnostic tests for *P. jirovecii* were performed according to the physicians’ decision to diagnose the causative pathogens of community-acquired or healthcare-associated pneumonia. PCP, assigned as PCP-confirmed inpatients, was strictly defined as the corresponding radiologic findings on chest X-ray or chest CT scan and positive results in *P. jirovecii* PCR, cytology, or histopathology from any respiratory specimens according to guidelines [45,46]. We thoroughly reviewed the chest radiologic findings of patients who had received diagnostic tests for *P. jirovecii*, assigned as PCP-suspected inpatients, to increase the power of analyses by verification of PCP diagnoses and exclusion of asymptomatic *P. jirovecii* colonization or unnecessary PCP tests. The affirmation of chest radiologic findings was performed using a text search of the radiologists’ readings with the keywords “pneumonia”, “consolidation”, and “infiltration” in the RDBMS [45,46].

After excluding 734 outpatients and 27 inpatients with unnecessary PCP tests, we finally selected the PCP-confirmed (*N* = 2922) among the PCP-suspected (*N* = 20,073) inpatients (Figure 1). We excluded all repeated results for PCP tests (*N* = 16,783) in each patient, and then, we included the first positive and negative results in patients with and without PCP among PCP-suspected patients, respectively (Figure 1). The index date for PCP diagnosis was defined as the date of the first positive and negative results in patients with and without PCP, respectively. We calculated the average numbers of observed PCP-confirmed cases per month in each year (between January and June, only for 2021) to perform the annual time-series analyses. The rate of PCP at certain time points was calculated using the following equation: (number of PCP-confirmed patients/number of PCP-suspected patients). The pre-COVID-19 and post-COVID-19 periods were defined as the periods before and after 1 January 2020. 

Data regarding age, sex, index date, ICU admission, all-cause of death during hospitalization, and immunocompromised status were collected from the RDBMS. We stated the immunocompromised conditions as higher risk morbidities for PCP for solid organ (SOT) or hematopoietic stem cell transplantation (HSCT), solid cancers, hematologic malignancies, human immunodeficiency virus (HIV)-1 infection, chronic lung diseases (chronic obstructive pulmonary disease, interstitial lung disease, and idiopathic pulmonary fibrosis), and corticosteroid therapy in this study [20,47]. High-dose and long-term corticosteroid therapy was defined as ≥20 mg/day of prednisone or equivalent when administered for ≥14 consecutive days [48]. There were no intra-hospital PCP outbreaks or large construction events that could cause air transmission during the study period in any hospital [20,26]. This study was approved by the united institutional review board sharing system of Yonsei University Medical Center at the Gangnam Severance Hospital (IRB No. 3-2021-0211) and Anam Hospital (IRB No. 2021AN0414). The requirement for Informed consent was waived.

### 2.2. Non-Pharmacological Interventions against the COVID-19 Pandemic Imposed by the Korean Government

After the first large outbreak in relation to religious gatherings in Dageu/Gyeongbuk province at the end of February 2020, the Korean government had executed the emergent enforced social distancing with measures to restrict the operation of multi-use facilities at risk of mass transmission, prohibition of gathering, call to actions for all citizens or for citizens in the workplace, and recommendation for mask wearing until early May. Since then, while the relaxed social distancing in life has been implemented, the second outbreak in August resulted in the stricter enforcement of the enhanced social distancing again (2 or 3 out of 4 levels). The government changed the social distancing to the lowest level 1 from mid-October and reorganized the levels in the form of precision quarantine with five steps from early November 2020. The number of COVID-19 patients confirmed by real-time reverse transcription PCR (RT-PCR) in South Korea was 300–800 per day due to clustered infection from December 2020 to June 2021, which subsequently increased to 2000 cases/day from July 2021 [49,50]. Since the highest surge in new cases in early December 2020, the government has been maintaining enhanced social distancing (2, 2.5, and 3 levels) to prohibit personal gatherings and businesses, such as restaurants, at night until the end of June (see the details for social distancing levels in Appendix A). The strong recommendation of mask wearing, except mesh or valve type masks and cloths or scarfs, turned into a compulsory fulfillment, as a fine penalty for not wearing a mask in the multi-use facilities was imposed from November 2020 [51,52,53,54].

### 2.3. Protective Policies for Prevention of Intra-Hospital SARS-CoV-2 Transmission

Four hospitals implemented the following policies to prevent nosocomial SARS-CoV-2 outbreak during the post-COVID-19 period: (1) daily web-based survey for COVID-19-associated symptoms or history of exposure in all HCWs, (2) exclusion from work for HCWs who had symptoms or history of contact or were an actively monitored subject by the local government, (3) prohibition of gathering/meeting in hospitals and of departure from abroad, (4) regular active surveillance for high-risk HCWs caring for suspected or confirmed patients and for caregivers or guardians staying for a long time, (5) prohibition of any family visit in ICU, (6) permission for one family visit with a negative SARS-CoV-2 PCR result in the general ward, (7) admission of patients after confirmation of a negative PCR test and isolation in a negative pressure ventilation room for patients referred from long-term care facilities or other hospitals until obtaining a negative PCR, (8) compulsory care for all outpatients with fever or respiratory symptoms or a history of close contact with suspected or confirmed COVID-19 patients in safety or respiratory clinics outside the hospital building, and (9) mandatory mask wearing by all HCWs and outpatients/visitors [55]. Active surveillance was performed along with collection of nasopharyngeal and oropharyngeal swabs for SARS-CoV-2 quantitative real-time RT-PCR tests targeting the *N*, RdRp/S, and E genes or ORF1ab of the RdRp and E genes.

### 2.4. P. Jirovecii PCR

The real-time PCR tests for *P. jirovecii* at Severance Hospital and Anam Hospital were performed using the AmpliSens^®^ *P. jirovecii*-FRT PCR kit (InterLabService Ltd., Moscow, Russia), which is a widely used and well-validated assay targeting the mitochondrial large subunit ribosomal RNA locus (*mtLSU* rRNA) [56], on a CFX96^™^ Real-time PCR Detection instrument (Bio-Rad, Hercules, CA, USA). Gangnam and Yongin Severance Hospital used the AmpliSens^®^ *P. jirovecii*-FRT PCR kit until October 2016 and, subsequently, the PCR test targeting *mtLSU* rRNA (Seegene Inc., Seoul, Korea) on a SEEAMP^™^ thermal cycler (Seegene) [57]. All real-time PCR tests for 40 cycles determined the positive, weak positive, and negative results as the cycle threshold (Ct) values of <35, 35–37, and >37, respectively. We defined the positive and weak positive result as the final positive test.

### 2.5. Statistical Analysis

Data are presented as the mean ± standard deviation or percentage. We used the independent *t*-test and χ^2^ test to compare the two groups. Data without a normal distribution were expressed as the median (interquartile range) and compared using the non-parametric Mann–Whitney U test between the groups. The statistical analyses for baseline characteristics were performed using SPSS software (version 25; IBM Corp., Armonk, NY, USA).

We applied the index date for the annual traditional and Bayesian structural time-series (BSTS) analyses to forecast the PCP cases in the post-COVID-19 period with the observed data (average of PCP cases per month in each year) from the pre-COVID-19 period and compare the monthly average number of forecasted and observed cases in the post-COVID-19 period. For the traditional time-series analyses, the exponential smoothing (ETS) and autoregressive integrated moving average (ARIMA) model for interrupted time-series analysis were performed using the Python programming language (version 3.9.6) with the Pandas library (version 1.3.0) and statsmodels package (version 0.12.2) [58,59]. The seasonal decomposition in the EST model by multiplicative error assumption was performed to obtain the trends of PCP-suspected and confirmed inpatients and PCP rates with quarterly or yearly frequency [59]. The residuals indicating the difference between the forecasted and observed data in the ARIMA model were expressed as estimates (standard error) (%). Graph visualization was performed using the Matplotlib library (version 3.4.2) in Python.

Additionally, we employed the Bayesian structural state-space model for time-series, which was proposed by Google Inc. in 2015 and has been implemented in public health research, using SAS version 9.4 and R language (version 4.1.0) with the CausalImpact package [60,61,62,63,64]. This Bayesian model determines the causal impact of a planned intervention by acquiring a counterfactual prediction in an artificial control of what would have taken place had this intervention not occurred [60,63]. The intervention in this study indicated that stringent nationwide NPIs arose due to the COVID-19 pandemic. A counter fact, which implied that PCP cases would have developed if the COVID-19 pandemic did not occur, was obtained from the true (observed) PCP cases during the pre- and post-COVID-19 periods. The causal impact of enforced national and hospital policies on PCP cases was estimated by calculating the pointwise (each year, between January and June only for 2021) and cumulative (the whole post-COVID-19 period) residuals, which were the distinctions between the overall observed and counterfactual (predicted) PCP cases in the post-COVID-19. The average absolute (observed–predicted) and relative ([observed–predicted]/predicted × 100) casual effects caused by the intervention (COVID-19 pandemic) were expressed as percentages and 95% counterfactual prediction credible intervals (CIs) [65]. A two-tailed *p*-value < 0.05 was considered statistically significant in all statistical analyses.

## 3. Results

### 3.1. Clinical Information of Total PCP-Confirmed Inpatients

Only three patients among 22 (pre-) and 5 (post-COVID-19) inpatients who were tested clinically with an unnecessary PCR without the suspicion of PCP had weak positive results in the pre-COVID-19 period. Two patients had chronic emphysema and idiopathic pulmonary fibrosis, and one patient was hospitalized for SOT without respiratory symptoms or abnormal lung parenchyma. The PCP rates were 14.6% (2922 PCP-confirmed patients/20,073 PCP-suspected patients), 15.2% (2163/14,192), and 12.9% (759/5881) during the total study duration and pre- and post-COVID-19 periods, respectively (Table 1 and Figure 1).

The mean age and percentage of male patients in the total PCP-confirmed inpatients were 66 years and 70.5% for all periods, respectively; 31.7%, 35.7%, and 61.2% of total PCP cases had a history of ICU care, all-cause mortality during admission at PCP diagnosis, and immunocompromised status, respectively. The PCP-confirmed patients during the post-COVID-19 period had a significantly higher mean age (*p* < 0.001) and percentage of corticosteroid therapy (*p* = 0.002) or solid cancers (*p* < 0.001) as well as a lower percentage of ICU care (*p* < 0.001) or SOT recipients (*p* = 0.012) than patients during pre-COVID-19 (Table 1). The means of quantitative Ct cycles in PCP-confirmed patients were similar between pre- and post-COVID-19 periods (30.9 ± 9.4 vs. 29.1 ± 10.5, *p* = 0.503). The percentages of weak positive with lower fungal load were only 8.1% and 7.6% for pre- and post-COVID-19, respectively (Table 1).

### 3.2. Trend Analysis and ARIMA Model to Compare the Observed and Forecasted PCP Cases in the Post-COVID-19

PCP most commonly occurred in June (mean, 43.9 cases and 16.9%) and May (43.0 and 16.5%) during the total study period. Additionally, the annual numbers of observed PCP cases were highest and lowest in 2019 (565) and 2015 (338), respectively (Appendix A). The plots of the observed PCP cases in both total and subgroups among PCP-suspected patients showed a diverse transition without obvious upward or downward change and seasonality in the pre- and post-COVID-19 (Figure 2 and Appendix A). Trend analysis through the ETS model revealed a steadily increasing pattern for the total number of PCP-suspected or -confirmed patients in both the quarterly and yearly frequency during the total study period but not for the PCP rates (Appendix A).

As our time-series data showed stationary characteristics without seasonality, which means that the mean, variance, and covariance of data were invariant to time, in the augmented Dickey–Fuller test (*p* < 0.001) [66], we applied the non-seasonal ARIMA (1, 0, 1) model using the following parameters, with the lowest Akaike Information Criterion value (388.7): (1) 1 of autoregression (p) from the autocorrelation function of residuals, (2) 0 of degree of differencing (integrated, d), and (3) 1 of size of the moving average window (q) from the partial autocorrelation function of residuals (Appendix A) [59,67,68]. As the observed PCP-confirmed cases did not exist for several months, and average numbers of observed PCP-confirmed cases per month in each year were very small in the pre- and post-COVID-19 periods, we did not perform the time-series analysis for HSCT recipients, chronic lung disease, and HIV-1-infected individuals in the ARIMA and BSTS model.

The plots and regression lines of the monthly observed PCP cases in total PCP-confirmed patients did not show a distinct and continuous pattern during the post-COVID-19 period, with the generally lowest cases in November and December 2020. In addition, the observed PCP cases were similar between the latest time point (June 2021) in the post-COVID-19 period and the latest month (December 2019) for pre-COVID-19 in the total patients (Appendix A). The ARIMA model did not reveal a significant difference between the observed and forecasted PCP cases in total PCP-confirmed inpatients during post-COVID-19 (7.4% of residual estimate, *p* = 0.765). In subgroup analyses, the observed PCP in patients with each immunocompromised condition were similar with the forecasted PCP cases for post-COVID-19. The observed and forecasted PCP cases in patients without immunocompromised conditions and critically ill patients receiving ICU care did not differ for post-COVID-19 (Table 2) 

### 3.3. Bayesian Structural Time-series Model to Compare Observed and Predicted PCP Cases in the Post-COVID-19

The monthly average numbers of observed PCP cases (47.3) in the total PCP-confirmed inpatients for post-COVID-19 was higher than that (36.1) for pre-COVID-19. However, the counterfactually predicted PCP cases (48.0, 95% CI: 33.7–55.9) for post-COVID-19 by the BSTS model were not significantly different from the observed PCP cases for post-COVID-19 in the total PCP-confirmed patients (absolute effect, −0.6%, 95% CI: [−10~9%] and relative effect, −1.3% [−20~19%], *p* = 0.450) (Table 3 and Figure 2a). In subgroup analyses, the predicted PCP cases in the patients with each immunocompromised condition, age group, and sex were not different from the observed PCP cases for post-COVID-19 (Table 3 and Figure 2). However, the PCP patients receiving ICU care (3.0% [1~6%] and 24.0% [5~44%], *p* = 0.012) had significantly higher monthly average numbers of observed PCP cases (15.7%) than the predicted cases (12.7%) for post-COVID-19 (Table 3 and Figure 2e). The cumulative residuals of PCP cases in patients with ICU care steadily increased with statistical significance in the post-COVID-19 period (Figure 2e).

## 4. Discussion

Along with the enforced infection control measures in the community setting, our strict restriction of hospital visits could influence the spread of *P. jirovecii* from guardians to inpatients, even though we did not perform active surveillance for *P. jirovecii* carriers in asymptomatic recurrent or long-term residents in the hospitals, in contrast to SARS-CoV-2. However, these time-series models did not reveal the protective effects of unprecedented stricter NPIs, particularly mandatory mask wearing and exhaustive entrance control in hospitals, except for contact isolation on PCP incidence, contrary to the post-COVID-19 epidemiologic change of influenza and CA-RVs in various countries and continents, which are in line with general expectations [11,12,13,14,15,18,19]. Our data showed a declining tendency of PCP in post-COVID-19, but the effect of enforced NPIs during the COVID-19 pandemic on occurrence of PCP was not confirmed by time-series models. We intuitively observed that the PCP cases declined slightly in the early period of post-COVID-19 (monthly average: 40 in 2020) and, afterward, increased from 2021 (47 between January 2021 and June 2021) (Appendix A). Long-term follow-up studies will be needed in the specific patients with a high risk for PCP.

In contrast to the declining pattern of PCP in HIV-1-infected individuals, the frequency of PCP in the developed countries has been increasing among the HIV-negative population with new risk groups, including hematologic malignancies, solid cancers, chronic lung diseases, and post-transplant status during the last two decades in the era of highly active anti-retroviral therapy against HIV-1 before the COVID-19 pandemic [27,69,70,71]. The number of PCP cases in patients not infected with HIV-1 increased from 4.4 to 6.3 cases per million between 2008 and 2012 from national hospital discharge records in Spain [70], cases increased from 157 in 2000 to 352 in 2010 from hospital episode statistics data in England [69], and cases increased from 25 in 2007 to 46 in 2017 from regional referral laboratory in Central Norway [71], like observed for our data (from 331 in 2015 to 465 in 2020 among HIV-negative patients). Our subgroups analyses included these risk population of PCP from the recent studies [27,69,70,71].

The main reason for the persistence of the PCP diagnosis during the COVID-19 pandemic would probably be associated with the much lower transmission probability and infectivity of *P. jirovecii* in the air compared to airborne-transmitted viruses (estimated basic reproduction number [R0] during the early stages of an epidemic or pandemic: 1.5–5.5 for SARS-CoV-2, 1.3–23.0 for influenza, 3.0 for RSV, and 3.7–57.0 for measles) [27,31,72,73,74,75,76]. Little is known about R0 for *P. jirovecii*. However, the fungal load (relatively high Ct values with ≥30 or small copies with 10–4000/µL in PCR tests) of *P. jirovecii* exhaled in the air is lower than with viral infections [31,34,36,37,41,42], and potentially, not all the asymptomatically acquitted *P. jirovecii* propagules can induce sustained PCP [20,27,31]. In addition, the airborne-transmission of *P. jirovecii* should require common contact with contaminated air, indicating that transmission is much more probable with people who meet frequently [27,31,42,77,78]. Reinforced social distancing and nationwide infection control measures inversely increased common close contact within the home [7]. Transmission of *P. jirovecii* would not be affected by preventing air transmission from the general population, but rather, it occurs within family members (for instance, between parents and children) under more sustained and close contact, which was not affected by strict NPIs during the COVID-19 pandemic.

It is recommended that the PCP-suspected and -confirmed patients should be hospitalized in a single room [30]. Our hospitals did not conduct contact isolation precautions, such as wearing gloves and gowns in specific single beds, hospital rooms, or wards as a cohort isolation for PCP inpatients. Additionally, the majority of patients receive hospital care in multiple occupations (2, 4, or 6 persons in one room) and open spaces, rather than a single isolation room in the ICU. Because contact isolation for PCP is not recommended, even in severely immunocompromised hosts [45,46], the structural vulnerabilities in inpatient beds might countervail the preventive roles of enforced NPIs for *P. jirovecii* acquisition.

We also need to consider all the non-COVID-19 income restriction policies in hospitals during the pandemic and the possible changes in the management of patients with non-COVID-19 diseases who visit the hospital during this period due to the care overload caused by enforced NPIs. During the pandemic, the Korean government has steadily implemented a reimbursement policy to compensate for the decline in hospitals’ revenue caused by strict restriction of visitors and treatment of COVID-19 patients in special negative pressure and closed wards. In addition, all Koreans are covered by the compulsory national health insurance policy. Furthermore, our hospitals increased staff and nurses dedicated to infection control measures, as well as critical care and/or the emergency room. Several of these reasons could make it less likely that patients infected by *P.jirovecii* will not be able to come to the hospital, even during COVID-19 pandemic.

The sentinel surveillance systems are being implemented for seasonal influenza or various communicable infectious diseases in several countries [79,80]. The pathogens with highly contagious and/or public risk should be totally monitored with mandatory reporting [81]. However, the surveillance systems also have the limitation of underreporting or a lapse of surveillance [82]. The exact incidence and prevalence of PCP are difficult to determine, because the large surveillance system for PCP is not available worldwide, regardless of the clinical burden and severity. These factors may make it hard to select a target group for our investigation. Our study should be interpreted in consideration of the following points as probable biases: (1) difficult diagnostic process for *P. jirovecii*, even in patients with symptomatic lung infiltration, (2) not controlling for the patients who had imaging compatible for PCP and for whom no *P. jirovecii* test was requested, (3) not conducting surveillance tests in asymptomatic individuals, (4) not all patients were hospitalized, even though our data had very low rate of PCP-confirmed (*N* = 26) and suspected (*N* = 660) outpatients in the total duration of 78 months, and (5) a lack of sophisticated analysis using quantitative PCR values to examine the transmission risk of *P. jirovecii* with a low concentration

However, this first attempt to analyze the potential influence of stricter NPIs during the COVID-19 pandemic on occurrence of PCP has some strengths: (1) meticulous forecasting and interpretation using different time-series prediction models reflecting time variation, particularly the Bayesian model inferring the causal impact of strict NPIs and/or the COVID-19 pandemic on PCP cases and (2) the exclusion of possible biases by the change of the total number of *P. jirovecii* tests, especially the underestimation of PCP diagnosis by the decreased numbers of tests or suspicious cases. In fact, the number of PCP-suspected patients increased during the post-COVID-19 period.

Our study has some limitations. First, a major hurdle was the convoluted selection of the exposed or target population to obtain an accurate incidence rate of PCP. Our study did not focus on a specific population with a higher risk for PCP. The total number of patients at admission during the study period may not be an appropriate parameter of the individuals at PCP risk because the majority of inpatients may be irrelevant to PCP risk. As PCP is not classified as a legal communicable disease in South Korea, we could not use the nationwide mandatory or sentinel infectious disease surveillance system. Second, this study could not perform the active surveillance to evaluate the *P. jirovecii* colonization in inpatients without suspicion of PCP or hospital visitors or HCWs. Third, PCP might be underdiagnosed in any particular period, because we could not consider all patients with interstitial pneumonia, as they did not have any tests for *P. jirovecii*. Fourth, we could not analyze the specific immunocompromised subgroups, because the PCP did not occur for several months during the pre-COVID-19 period, and time-series analyses might generate unreliable results arising from random fluctuations. Fifth, our data did not demonstrate an association between NPIs and PCP in provinces other than Seoul and Gyeonggi-do, which have the largest number of SARS-CoV-2-confirmed patients and maintain the strongest social distancing measures and local government policies in South Korea [83]. Lastly, even though the overall adherence for mask wearing and control measures may be constantly high in the general population in public areas, we could not measure the exact adherence around the clock in hospital beds.

## 5. Conclusions

The unprecedented infection control measures against the COVID-19 pandemic were not associated with mitigation of PCP, unlike other respiratory viruses, including seasonal influenza transmitted by aerosols or droplets. Further large studies are needed to uncover detailed epidemiologic evidence and the pathogenic mechanisms of PCP development.

## Figures and Tables

**Figure 1 jof-07-00990-f001:**
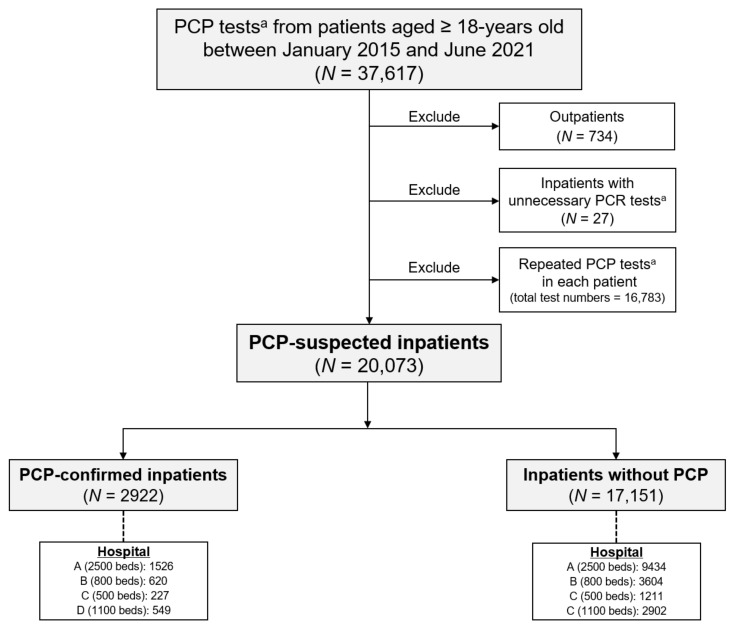
Schematic presentation to select the PCP-suspected and -confirmed inpatients. PCP tests^a^ include PCR and cytology in the sputum, bronchial washing, and BAL fluid, as well as histopathology in lung or bronchial tissue. A, B, C, and D indicate the Severance, Gangnam Severance, Yongin Severance, and Anam hospitals, respectively. Abbreviations: BAL, bronchoalveolar lavage; PCP, *P. jirovecii* pneumonia; PCR, polymerase chain reaction.

**Figure 2 jof-07-00990-f002:**
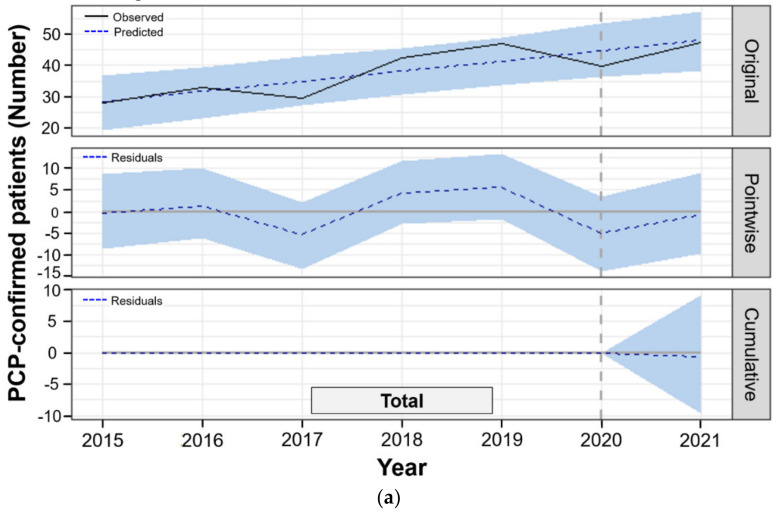
Channing pattern of the observed and predicted PCP-confirmed inpatients in the pre- and post-COVID-19 periods by the Bayesian structural time-series model. (**a**) All PCP-confirmed inpatients; (**b**) age; (**c**) sex; (**d**) immunocompromised conditions; (**e**) ICU care. The number of PCP-confirmed patients (Y-axis) indicated the average of observed PCP cases per month in each year. Gray dotted vertical lines (January 2020) indicate that the intervention of enforced NPIs arose from the COVID-19 pandemic in this study. The sky blue zone indicates the values between the upper and lower 95% CIs. Abbreviations: CI, credible interval; ICU, intensive care unit; ImmC, immunocompromised conditions; NPI, non-pharmacological intervention; PCP, *P. jirovecii* pneumonia; SOT, solid organ transplantation.

**Table 1 jof-07-00990-t001:** Clinical information of PCP-confirmed inpatients between January 2015 and June 2021.

	Total Duration (*N* = 2922)	Pre-COVID-19 (*N* = 2163)	Post-COVID-19 (*N* = 759)	*p*-Value ^a^
Age, years	65.7 ± 13.3	65.0 ± 13.4	67.5 ± 12.9	<0.001
<65	1189 (40.7)	921 (42.6)	268 (35.3)	
≥65	1733 (59.3)	1242 (57.4)	491 (64.7)	
Sex, male	2061 (70.5)	1520 (70.3)	541 (71.3)	0.257
ICU care at admission	926 (31.7)	707 (32.7)	219 (28.9)	<0.001
All-cause death at admission	1044 (35.7)	780 (36.1)	264 (34.8)	0.128
Hospital stay, days	19 (10–36)	18 (10-36)	19 (11–37)	0.753
PCP diagnosis				
PCR				
Sputum	2492 (85.3)	1884 (87.1)	608 (80.1)	<0.001
Bronchial washing	136 (4.7)	66 (3.0)	70 (9.2)	0.176
BAL fluid	214 (7.3)	166 (7.7)	48 (6.4)	0.859
Cytology	59 (2.0)	30 (1.4)	29 (3.8)	0.357
Histopathology	21 (0.7)	17 (0.8)	4 (0.5)	0.795
Quantitative cycles in real-time PCR tests, Ct values	29.2 ± 11.2	30.9 ± 9.4	29.1 ± 10.5	0.503
Positive ^b^	2688 (92.0)	1987 (91.9)	701 (92.4)	0.718
Weak positive ^c^	234 (8.0)	176 (8.1)	58 (7.6)	0.652
Immunocompromised conditions	1789 (61.2)	1329 (61.4)	460 (60.6)	0.935
SOT recipients	260 (8.9)	220 (10.2)	40 (5.3)	0.012
HSCT recipients	35 (1.2)	25 (1.2)	10 (1.3)	0.912
Solid cancers	738 (25.3)	526 (24.3)	216 (28.5)	<0.001
Hematologic malignancies	335 (11.5)	252 (11.7)	83 (10.9)	0.752
HIV-1 infection	62 (2.1)	52 (2.4)	10 (1.3)	0.376
Chronic lung diseases	53 (1.8)	47 (2.2)	6 (0.8)	0.121
High-dose and long-term corticosteroid therapy	306 (10.5)	207 (9.6)	95 (12.5)	0.002

Data are expressed as the mean ± standard deviation, median (interquartile range), or number (percentage). ^a^ Comparison between pre- and post-COVID-19 periods. ^b,c^ Indicate the Ct values of <35 and 35–37, respectively. Abbreviations: BAL, bronchoalveolar lavage; Ct, cycle threshold; HIV, human immunodeficiency virus; HSCT, hematopoietic stem cell transplantation; ICU, intensive care unit; PCP, *P. jirovecii* pneumonia; PCR, polymerase chain reaction; SOT, solid organ transplantation.

**Table 2 jof-07-00990-t002:** Autoregressive integrated moving average analysis to compare the observed and forecasted PCP cases in the post-COVID-19 pandemic period.

Characteristics	Estimate (SE) (%)	*p*-Value
Total PCP-confirmed inpatients	7.42 (22.64)	0.765
Age (years old)		
<65	0.18 (7.07)	0.981
≥65	7.27 (15.7)	0.675
Sex		
Male	5.38 (6.44)	0.465
Female	2.03 (6.64)	0.780
Immunocompromised conditions		
No	4.04 (7.13)	0.610
Yes	3.79 (6.73)	0.613
SOT recipients	−1.25 (0.96)	0.282
Solid cancers	2.92 (2.65)	0.351
Hematologic malignancies	0.61 (2.99)	0.851
High-dose and long-term corticosteroid therapy	1.70 (1.53)	0.346
ICU care at admission		
No	6.13 (6.60)	0.421
Yes	1.28 (4.21)	0.782

As the average numbers of observed PCP-confirmed cases per month in each year were very small in the pre- and post-COVID-19 periods, we did not include data from HSCT recipients, chronic lung disease, and HIV-1-infected individuals. Abbreviations: ICU, intensive care unit; PCP, *P. jirovecii* pneumonia; SE, standard error; SOT, solid organ transplantation.

**Table 3 jof-07-00990-t003:** Bayesian structural time-series model to compare the observed and counterfactually predicted PCP cases in the post-COVID-19 pandemic period.

Characteristics	Pre- COVID-19	Bayesian Structural Time-Series Model				
		Post-COVID-19				
	Observed PCP Cases (No.) ^a^	Observed PCP Cases (No.) ^a^	Predicted PCP Cases (No.) ^a^	Absolute Average Effect (%) (95% CI)	Relative Average Effect (%) (95% CI)	*p* -Value
Total PCP-confirmed inpatients	36.1	47.3	48.0	−0.6 (−9.6~9.1)	−1.3 (−20.0~19.0)	0.450
Age (years old)						
<65	15.4	17.3	17.1	0.2 (−2.9~3.6)	1.0 (−17.0~21.0)	0.448
≥65	20.7	30.0	30.8	−0.8 (−7.9~5.8)	−2.7 (−26.0~19.0)	0.392
Sex						
Male	25.3	33.0	34.6	−1.6 (−8.3~5.6)	−4.7 (−24.0~16.0)	0.319
Female	10.7	14.3	13.3	1.0 (−1.6~3.9)	7.7 (−12.0~29.0)	0.221
Immunocompromised conditions						
No	13.9	20.2	19.1	1.0 (−3.1~5.5)	5.4 (−16.0~29.0)	0.308
Yes	22.2	27.2	26.1	1.1 (−1.4~3.7)	4.0 (−5.4~14.0)	0.202
SOT recipients	3.8	5.7	4.9	0.7 (−0.1~1.7)	15.0 (−2.0~33.0)	0.072
Solid cancers	9.3	12.5	11.8	0.7 (−0.7~2.3)	6.0 (−6.0~20.0)	0.166
Hematologic malignancies	4.2	5.2	5.5	−0.3 (−1.8~1.3)	−5.0 (−32.0~24.0)	0.347
High-dose and long-term corticosteroid therapy	3.7	2.2	3.1	−0.9 (−1.9~0.2)	−30.0 (−63.0~5.0)	0.057
ICU care at admission						
No	24.3	31.7	35.3	−3.6 (−10.7~4.1)	−10.0 (−30.0~12.0)	0.170
Yes	11.8	15.7	12.7	3.0 (0.6~5.6)	24.0 (4.5~44.0)	0.012

^a^ Indicates the monthly average numbers of PCP-confirmed cases in the pre- and post-COVID-19 periods. As the average of observed PCP cases per month in each year were very small in the pre- and post-COVID-19 period, we did not include data from HSCT recipients, chronic lung disease, and HIV-1-infected individuals. Abbreviations: CI, credible interval; ICU, intensive care unit; No., number; PCP, *P. jirovecii* pneumonia; SOT, solid organ transplantation.

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
