# Peer review of "No Change of *Pneumocystis jirovecii* Pneumonia after the COVID-19 Pandemic: Multicenter Time-Series Analyses"

_jof, 2021, doi:10.3390/jof7110990_

Round 1

Reviewer 1 Report

The article addresses the effect of non-pharmacological interventions used against COVID-19 pandemic on Pneumocystis Pneumonia incidence. The idea is original, and the article is clear and well-written.  The main limitation of this study, that is recognized by the authors, is the use of number of suspected cases instead of individuals at Pneumocystis pneumonia risk to obtain the incidence rate of Pneumocystis pneumonia.

Only a minor issue.

In table 4, the data about hematopoietic stem cell transplantation is missing but HSCT (hematopoietic stem cell transplantation) appears as a footnote to the table

Author Response

The article addresses the effect of non-pharmacological interventions used against COVID-19 pandemic on Pneumocystis Pneumonia incidence. The idea is original, and the article is clear and well-written. The main limitation of this study, that is recognized by the authors, is the use of number of suspected cases instead of individuals at Pneumocystis pneumonia risk to obtain the incidence rate of Pneumocystis pneumonia.

→ We do appreciate the detailed review and consideration for publication. According to the Reviewer #2’s comment, we modified the entire time-series data using average numbers of observed PCP-confirmed patients (cases) per month in each year instead of PCP rates in the further revised manuscript. Please see our answers and changes for first comment by the Reviewer #2. In spite of this methodological changes, there was no changes in the overall data and conclusion. We think that the alteration by the Reviewer #2’s comment could weaken the limitation for the incidence rate of PCP. However, we maintained the discussion as our limitation in the further revised manuscript.

  • Our study has some limitations. First, a major hurdle was the convoluted selection of the exposed or target population to obtain an accurate incidence rate of PCP. Our study did not focus on a specific population with a higher risk for PCP. The total number of patients at admission during the study period may not be an appropriate parameter as the individuals at PCP risk because the majority of inpatients may be irrelevant to PCP risk. As PCP is not classified as a legal communicable disease in South Korea, we could not use the nationwide mandatory or sentinel infectious disease surveillance system.

Only a minor issue.

In table 4, the data about hematopoietic stem cell transplantation is missing but HSCT (hematopoietic stem cell transplantation) appears as a footnote to the table

→ We deleted the abbreviations for HSCT and HIV in footnote of the revised Table 3, Table 4, and Figure 2.

Reviewer 2 Report

I read with interest the article by Kim and colleagues entitled “No Change of Pneumocystis jirovecii Pneumonia after the COVID-19 Pandemic: Multicenter Time-series Analyses” submitted for publication in Journal of Fungi

In this interesting study the authors conducted a multicenter time-series analyses in Korea to assess changes in the P. jirovecii pneumonia (PcP) rates in a hospital setting before and after the COVID-19 pandemic. They conclude that strict non-pharmacological interventions to prevent transmission of SARS-CoV2 infection do not affect PcP rates.

In my opinion, there are important methodological and epidemiological questions that have not been adequately considered in the study and that limit the scope of the conclusions.

First, the time-series analysis should be performed for confirmed PcP cases and not for cases of suspected PCP, most of which are ruled out by microbiological techniques (PCR and / or staining). Since PcP is a serious hospital-managemend disease, in this way we could really answer the study question: has the incidence of PcP changed during the COVID-19 pandemic?

In this sense, the definition of the "rate of PcP" that is used in the article is wrong, since it is really “the confirmation rate in cases of suspected PcP”. The rate of PcP is the number of incident cases of PcP.

The analysis on suspected cases and the positivity rate, although it is interesting, is a "construct" probably for increasing the "n" to increase and in this way the power of the analyses and, in my opinion, it should be passed to supplementary data. The authors must modify the entire time-series analyses and focus on the monthly or annual confirmed PcP cases. These last are not even indicated in the article, although they can be deduced from table 2 of the supplementary material: 338 in 2015, 396 in 2016, 354 in 2017, 510 in 2018, 565 in 2019, 475 in 2020 and 284 in the first half of 2021

Second, from an epidemiological point of view the authors only consider the impact of policies to prevent SARS-CoV-2 outbreak during the post-COVID-19 period, but they do not consider at all the non-COVID income restriction policies in hospitals during the pandemic and the possible changes in the management of patients with non-COVID diseases who enter this period due to the care overload caused by the epidemic. Both factors should be described and discussed in the manuscript.

Finally, I believe that the authors should also consider the secular trends of changes in the incidence of PcP that have been observed in developed countries in the last two decades before the COVID outbreak, and discuss their findings comparatively with the studies in which they are analyzed (mainly two: Maini et al. PMID: 23622345 and, Pereira-Díaz et al . PMID: 31637227).

Author Response

1. First, the time-series analysis should be performed for confirmed PcP cases and not for cases of suspected PCP, most of which are ruled out by microbiological techniques (PCR and/or staining). Since PcP is a serious hospital-managed disease, in this way we could really answer the study question: has the incidence of PcP changed during the COVID-19 pandemic? In this sense, the definition of the "rate of PcP" that is used in the article is wrong, since it is really “the confirmation rate in cases of suspected PcP”. The rate of PcP is the number of incident cases of PcP. The analysis on suspected cases and the positivity rate, although it is interesting, is a "construct" probably for increasing the "n" to increase and in this way the power of the analyses and, in my opinion, it should be passed to supplementary data. The authors must modify the entire time-series analyses and focus on the monthly or annual confirmed PcP cases. These last are not even indicated in the article, although they can be deduced from table 2 of the supplementary material: 338 in 2015, 396 in 2016, 354 in 2017, 510 in 2018, 565 in 2019, 475 in 2020 and 284 in the first half of 2021

→ We do appreciate the detailed review and consideration for publication. We totally agree with this point. We changed the entire data (tables, figures, and main text) of time-series analyses (ARIMA and Bayesian structural time-series model) using the annual number [which was calculated by average of observed PCP cases per month in each year, especially, between January and June only for 2021] of PCP-confirmed patients (cases) instead of PCP rates in the further revised manuscript according to this comment.

→ We included the only annual time-series analyses (but not monthly) using the autoregressive integrated moving average and Bayesian structural time-series model of the revised manuscript, because the observed PCP-confirmed cases did not exist for several months and the average numbers of observed PCP-confirmed cases per month in each year were very small in the pre- and post-COVID-19 period among HSCT recipients, chronic lung disease, and HIV-1-infected individuals subgroups. In addition, we cannot perform the Bayesian structural time-series model using data from only one post-intervention period, that is, data from only one year (2020).

→ Because of these reasons, we obtained the average of the observed PCP cases per month in each year (especially, between January and June only for 2021) and predicted PCP cases in the post-COVID-19 period using the Bayesian structural time-series model. By doing this, we can use the observed PCP cases between Jan. 2021 and June 2021 in the time-series analysis.

→ We added the methodological details in the ‘Method’ section.

  • [1. Study design and data collection]: We calculated the average numbers of observed PCP-confirmed cases per month in each year (between January and June only for 2021) to perform the annual time-series analyses.
  • [5. Statistical analysis]: We applied the index date for the annual traditional and Bayesian structural time-series analyses to forecast the PCP cases in the post-COVID-19 period with the observed data (average of PCP cases per month in each year) from the pre-COVID-19 period and compare the monthly average number of forecasted and observed cases in the post-COVID-19 period. The causal impact of enforced national and hospital policies on PCP cases was estimated by calculating the pointwise (each year, between January and June only for 2021) and cumulative (the whole post-COVID-19 period) residuals, which was the distinction between the overall observed and counterfactual (predicted) PCP cases in the post-COVID-19.

→ Additionally, we performed the ARIMA model using the average of PCP cases per month in each year using only one data from post-intervention 2020 (from Jan. and Dec.). As shown in the table below, results were same with the Table 2.

Estimated difference(SE)

p-value

Total

-11.764 (340.592)

0.9756

Immunocompromised conditions, No

5.488 (279.665)

0.9861

Immunocompromised conditions, Yes

25.775 (114.464)

0.8428

 SOT recipients

-16.741 (34.968)

0.6793

 Hematologic malignancies

2.757 (49.836)

0.9609

Solid cancer

37.735 (20.892)

0.2126

Steroid use

18.744 (6.797)

0.1102

ICU, No

-57.058 (377.385)

0.8937

ICU, Yes

-14.879 (65.681)

0.8418

Age <65

-40.581 (144.281)

0.8049

Age ≥65

30.708( 221.202)

0.9023

Male

-0.389 (253.025)

0.9989

Female

3.992 (113.172)

0.9751

→ We believe the annual time-series analyses guarantee and strengthen the reliability of data analyses in our study. In spite of this statistically methodological changes, there was no changes in the overall data and conclusion.

→ Please see the revised Table 2 and 3 as well as Figure 2. With these changes, we modified the main text of “Abstract’, ‘Method’, ‘Result’, and “Discussion’ section in the revised manuscript. We marked up all changes using the “Track Changes” function in the manuscript.

→ According to the comment, we moved the figure 2 (PCP rates, ETS model, trends) into the Supplementary Figure 2

2. Second, from an epidemiological point of view the authors only consider the impact of policies to prevent SARS-CoV-2 outbreak during the post-COVID-19 period, but they do not consider at all the non-COVID income restriction policies in hospitals during the pandemic and the possible changes in the management of patients with non-COVID diseases who enter this period due to the care overload caused by the epidemic. Both factors should be described and discussed in the manuscript.

→ According to this comment, we inserted the below sentences.

  • [Discussion]: We also need to consider all the non-COVID-19 income restriction policies in hospitals during the pandemic and the possible changes in the management of patients with non-COVID-19 diseases who visit the hospital during this period due to the care overload caused by the enforced NPIs. During the pandemic, the Korean government has steadily implemented a reimbursement policy to compensate for the decline in hospitals' revenue caused by strict restriction of visitors and treatment of COVID-19 patients in special negative pressure and closed wards. In addition, all Koreans are covered by the compulsory national health insurance policy. And, our hospitals increased staff and nurses dedicated to infection control measures as well as critical care and/or emergency room. Several of these reasons could make it less likely that patients infected by jirovecii will not be able to come to the hospital even during COVID-19 pandemic.

3. Finally, I believe that the authors should also consider the secular trends of changes in the incidence of PcP that have been observed in developed countries in the last two decades before the COVID outbreak, and discuss their findings comparatively with the studies in which they are analyzed (mainly two: Maini et al. PMID: 23622345 and, Pereira-Díaz et al . PMID: 31637227).

→ According to the comment, we added the discussion about general changes of PcP in the last two decades before the COVID-19 pandemic with the new references.

  • [Discussion]: In contrast to the declining pattern of PCP in HIV-1-infected individuals, the frequency of PCP in the developed countries has been increasing among HIV-negative population with new risk groups, including hematologic malignancies, solid cancers, chronic lung diseases, and post-transplant status, during the last two decades in the era of highly active anti-retroviral therapy against HIV-1 before the COVID-19 pandemic [27,65-67]. The number of PCP cases in patients not infected HIV-1 increased from 4.4 to 6.3 cases per million between 2008 and 2012 from national hospital discharge records in Spain [66], cases from 157 in 2000 and 352 in 2010 from hospital episode statistics data in England [65], and cases from 25 in 2007 to 46 in 2017 from regional referral laboratory in Central Norway [67], like as our data (from 331 in 2015 to 465 in 2020 among HIV-negative patients). Our subgroups analyses included these risk population of PCP from the recent studies [27,65-67].
  • [New references]:

65. Maini, R.; Henderson, K.L.; Sheridan, E.A.; Lamagni, T.; Nichols, G.; Delpech, V.; Phin, N. Increasing pneumocystis pneumonia, england, uk, 2000-2010. Emerg Infect Dis 2013, 19, 386-392.

66. Pereira-Díaz, E.; Moreno-Verdejo, F.; de la Horra, C.; Guerrero, J.A.; Calderón, E.J.; Medrano, F.J. Changing trends in the epidemiology and risk factors of pneumocystis pneumonia in spain. Front Public Health 2019, 7, 275.

67. Grønseth, S.; Rogne, T.; Hannula, R.; Åsvold, B.O.; Afset, J.E.; Damås, J.K. Epidemiological and clinical characteristics of immunocompromised patients infected with pneumocystis jirovecii in a twelve-year retrospective study from norway. BMC Infect Dis 2021, 21, 659.

Round 2

Reviewer 2 Report

No further comments from  my side. Thanks for having considered my suggestions.

This manuscript is a resubmission of an earlier submission. The following is a list of the peer review reports and author responses from that submission.

Round 1

Reviewer 1 Report

General comments:

This is an interesting and well-written article.  It address to assess effect of non-pharmacological interventions used against COVID-19 pandemic on Pneumocystis Pneumonia incidence. The idea is original. The work objectives are clearly exposed and justified in the introduction. Methods and results are clearly exposed. Discussion is correct and tables and figures are informative.

Specific comments:

Title

As there is no change in incidence rate, I think it is better: ‘Pneumocystis jirovecii Pneumonia after the COVID-19 Pandemic: Multicenter Time-series Analyses’

Introduction 

(line 67-68) ‘Even though P. jirovecii pneumonia (PCP) can occur after environmental acquisition’. I think that this phase is not correct because Pneumocystis infection from environmental source is not demonstrate. However, if the authors know any evidence have to cite the article. The reference 18 that they used is only a review.

Materials and Methods

The study include patients between January 1, 2015 and June 30, 2021. Information about when and what non-pharmacological interventions against COVID-19 pandemic Korean government imposed should include in this section.

Results

Table 1

After COVID-19 PcP-Suspected inpatients decrease among immunocompromised diaseases, SOT recipients and Hematologic malignancies and have no changes among HSCT recipients and HIV-1 infection. In which group was the frequency of PcP- suspected inpatients increased? In non-immunosuppressed patients?

If it is the case, the probability of confirm a PcP is lower and the rate of PCP-suspected and confirmed inpatients should be also lower. This point should consider in the discussion.

Table 3

Relative average effect of 222.2 (49.0~373.0) in HSCT patients is right? Please, confirm it.

Discussion

Some articles have shown the risk of Pneumocystis infection among household. 

  • Morilla R, et al. Pneumocystis jirovecii among patients with cystic fibrosis and their household members. Med Mycol. 2021 Mar 10:myab010. doi: 10.1093/mmy/myab010.
  • Rivero L, et al. Pneumocystis jirovecii transmission from immunocompetent carriers to infant.  Emerg Infect Dis. 2008 ;14: 1116-8. doi: 10.3201/eid1407.071431.

Intra-family transmission of P. jirovecii could be considered in the discussion as one possible cause to explain the no mitigation of PcP incidence with the measures against the COVID-19 pandemic because face masks and social distancing do not apply in family environment.

Conclusion

(line 375-377) ‘Taken together our time-series analyses suggest that P. jirovecii may not be mainly acquired by airborne person-to-person transmission’.

I think that based exclusively on the results of this study is unwise to assume this affirmation.

Author Response

  1. Title: As there is no change in incidence rate, I think it is better: ‘Pneumocystis jiroveciiPneumonia after the COVID-19 Pandemic: Multicenter Time-series Analyses’

→ We fully consent to this point. Along with the Reviewer #2’s comment (“Since no change was observed, say “no change” in the title”.), we finally revised the tile as the following sentence.

  • Title: No Change of Pneumocystis jiroveciiPneumonia after the COVID-19 Pandemic: Multicenter Time-series Analyses

  1. Introduction: (line 67-68) ‘Even though P. jiroveciipneumonia (PCP) can occur after environmental acquisition’. I think that this phase is not correct because Pneumocystisinfection from environmental source is not demonstrate. However, if the authors know any evidence have to cite the article. The reference 18 that they used is only a review.

→ We totally agree with this comment. According to the reference 18 [NEJM review article: N Engl J Med 2004, 350, 2487-2498], the authors mentioned the following sentence: “Now there is evidence that person-to-person transmission is the most likely mode of acquiring new infections, although acquisition from environmental sources may also occur.29” (Reference 29: Morris A, Beard CB, Huang L. Update on the epidemiology and transmission of Pneumocystis carinii. Microbes Infect 2002;4:95-103.)

In reference 29 (Microbes Infect 2002;4:95-103.), the authors presented the various experimental data showing environmental sources of P. jirovecii including pond water, rural air and certain soil (association with camping, gardening or hiking than matched controls). Please see the below table 2 (Evidence for active acquisition of infection: environmental exposure).

→ In addition, Kpandja Djawe et al. reported the paper of “Environmental Risk Factors for Pneumocystis Pneumonia Hospitalizations in HIV Patients” in 2013 (Clinical Infectious Diseases 2013;56(1):74–8). They identified both climatological and air pollution constituents as independent risk factors for hospitalization of HIV-positive patients with PCP in San Francisco. This study shows that among climatic and ambient air pollutant constituents, temperature and SO2, present at high levels immediately (0–3 days) prior to admission appeared to be associated with the likelihood of being admitted with PCP among HIV-positive patients living in San Francisco.)

Figure 1. Total number of Pneumocystis pneumonia admissions by season and mean temperature. Abbreviation: PcP, Pneumocystis pneumonia. → Similar with our data, they also reported the relatively higher number of PCP admission during summer (relation to mean temaperature)

Along with these several studies, we carefully thought the phrase of “Even though P. jirovecii pneumonia (PCP) can occur after environmental acquisition” do not have the scientific evidences, nor very inconsistent (wrong) expression. The above previous data could enough support the phrase of “‘Even though P. jirovecii pneumonia (PCP) can occur after environmental acquisition”

→ We changed the word of “can” into “may” in the revised manuscript and added the various references directly showing the evidences for Pneumocystis environmental sources.

  • [ Introduction: Line - ]: Even though P. jirovecii pneumonia (PCP) may occur after environmental [18-24],

  • [New references]
  1. 19. Morris, A.; Beard, C.B.; Huang, L. Update on the epidemiology and transmission of pneumocystis carinii. Microbes Infect 2002, 4, 95-103.
  2. 20. Casanova-Cardiel, L.; Leibowitz, M.J. Presence of pneumocystis carinii DNA in pond water. J Eukaryot Microbiol 1997, 44, 28s.
  3. 21. Kaneshiro, E.S.; Maiorano, J.N. Survival and infectivity of pneumocystis carinii outside the mammalian host. J Eukaryot Microbiol 1996, 43, 35s.
  4. 22. Navin, T.R.; Rimland, D.; Lennox, J.L.; Jernigan, J.; Cetron, M.; Hightower, A.; Roberts, J.M.; Kaplan, J.E. Risk factors for community-acquired pneumonia among persons infected with human immunodeficiency virus. J Infect Dis 2000, 181, 158-164.
  5. 23. Dohn, M.N.; White, M.L.; Vigdorth, E.M.; Ralph Buncher, C.; Hertzberg, V.S.; Baughman, R.P.; George Smulian, A.; Walzer, P.D. Geographic clustering of pneumocystis carinii pneumonia in patients with hiv infection. Am J Respir Crit Care Med 2000, 162, 1617-1621.
  6. 24. Djawe, K.; Levin, L.; Swartzman, A.; Fong, S.; Roth, B.; Subramanian, A.; Grieco, K.; Jarlsberg, L.; Miller, R.F.; Huang, L. et al. Environmental risk factors for pneumocystis pneumonia hospitalizations in hiv patients. Clin Infect Dis 2013, 56, 74-81.

  1. Materials and Methods: The study includes patients between January 1, 2015 and June 30, 2021. Information about when and what non-pharmacological interventions against COVID-19 pandemic Korean government imposed should include in this section.

→ We do appreciate for this important information. According to this comment, we newly inserted the information of NPIs (non-pharmacological interventions) imposed by South Korean government in the revised manuscript (“Materials and Methods” section). Also, we provided the detailed contents of the NPIs including social distancing in the Supplementary Table 1. According to the reviewer #2’s comment (“Introduction: The number of COVID-19 patients confirmed by real-time reverse transcription polymerase chain…. such as restaurants, at night. Q: This paragraph is interesting but not directly linked to the present study and could be shortened). we moved the whole paragraph into the “Materials and Methods” section and included the detailed description about NPIs in the “Materials and Methods” section.

  • [ Materials and Methods - 2.2 Non-pharmacological interventions against COVID-19 pandemic imposed by the Korean government: Line - ]: After the first large outbreak in relation to religious gatherings in Dageu/Gyeongbuk province at the end of February 2020, the Korean government had executed the emergent enforced social distancing with measures to restrict the operation of multi-use facilities at risk of mass transmission, prohibition of gathering, call to actions for all citizens or for citizens in the workplace, and recommendation for mask wearing until early May. Since then, while the relaxed social distancing in life has been implemented, the second outbreak in August resulted in the stricter enforcement of the enhanced social distancing again (2 or 3 out of 4 levels). The government changed the social distancing to the lowest level 1 from mid-October and reorganized the levels in the form of precision quarantine with five steps from early November 2020. The number of COVID-19 patients confirmed by real-time reverse transcription PCR in South Korea was 300-800 per day due to clustered infection from December 2020 to June 2021, which subsequently increased to 2,000 cases/day from July 2021 (WHO COVID-19 Dashboard, https://covid19.who.int/region/wpro/country/kr, Korean Central Disease Control Headquarters, http://ncov.mohw.go.kr/en/). Since the highest surge in new cases in early December 2020, the government has been maintaining the enhanced social distancing (2, 2.5, and 3 levels) to prohibit personal gatherings and businesses, such as restaurants, at night. until the end of June (See the details for social distancing levels in Supplementary Table 1). The strong recommendation of mask wearing, except mesh or valve type mask and cloths or scarfs, turned into compulsory fulfillment as a fine penalty for not wearing a mask in the multi-use facilities had been imposed from November 2020 [53-56].

  • [New references]:
  1. Huh, K.; Jung, J.; Hong, J.; Kim, M.; Ahn, J.G.; Kim, J.H.; Kang, J.M. Impact of nonpharmaceutical interventions on the incidence of respiratory infections during the coronavirus disease 2019 (covid-19) outbreak in korea: A nationwide surveillance study. Clin Infect Dis 2021, 72, e184-e191.
  2. Korea Centers for Disease Control and Prevention. Enhanced social distancing campaign. Available online: http://ncov.mohw.go.kr/searchBoardView.do?brdId=3&brdGubun=32&dataGubun=321&ncvContSeq=1497 (accessed on September 13 2021).
  3. Central Quarantine and Countermeasure Headquarters. The covid-19 guidelines of the republic of korea for distancing in daily life. Available online: http://ncov.mohw.go.kr/en/guidelineView.do?brdId=18&brdGubun=181&dataGubun=&ncvContSeq=2763&contSeq=2763&board_id=&gubun=# (accessed on September 13 2021).
  4. Central Quarantine and Countermeasure Headquarters. Overview of social distancing systemoverview of social distancing system. Available online: http://ncov.mohw.go.kr/en/socdisBoardView.do?brdId=19&brdGubun=191&dataGubun=191&ncvContSeq=&contSeq=&board_id=&gubun= (accessed on September 13 2021).

Supplementary table 1. Non-pharmacological interventions including social distancing imposed and controlled by Korean government (Central Quarantine and Countermeasure Headquarters) during COVID-19 pandemic

Periods

Social distancing system

January 2020

~ June 2020

Enhanced social distancing

l Stay at home as much as possible.

l Call to actions for all citizens

·      Cancel or postpone all non-essential gathering, dining out, social events, and travel plans.

·      Avoid leaving home except to purchase necessities, to get medical care, or to go to work.

·      Avoid handshakes and other forms of physical contact. Keep a 2-meter distance from each other.

·      Wash your hands, cover up your sneezes/coughs, and generally. Maintain strict personal hygiene.

·      Disinfect and ventilate your space every day.

·      If you have fever, cough, sore throat, or other respiratory symptoms, do not go to work. Stay home and get sufficient rest.

l Call to actions for citizens in the workplace

·      Wash your hands thoroughly with soap under running water.

·      Use your own personal cups and utensils.

·      Refrain from using changing rooms, indoor break rooms, and other public areas.

·      Avoid handshakes and other forms of physical contact. Keep a 2-meter distance from each other.

·      When eating meals together, maintain a distance and avoid sitting face to face.

·      Return home directly after leaving work.

l It is recommended to wear a mask in the following cases.

·      Carrying for a COVID-19 patient

·      Having respiratory symptoms such as cough, sneezing, sputum, rhinorrhea, sore throat

·      Visiting medical institutions, pharmacies, seniors, disabled people, etc.

·      Working in a profession that requires contact with many people

·      When individuals with poor health or underlying medical conditions contact with other people within 2-meter in a poorly ventilated space

·      When using indoor multi-use facilities or it is not possible to keep a distance of 2-meter outdoors

Social distancing levels

July

2020

~ October 2020

1

2

3

4

Definition

Maintain a state of continuous restraint

Regional epidemic/People restrictions

Provincial epidemic

/prohibition gathering

Nationwide pandemic/No going out

Criteria

< 1 casesa/100,000

 1 ≤ casesa/100,000 < 2

2 ≤ casesa/100,000 < 4

≥ 4 casesa/100,000

Personal gathering

Compliance with quarantine rules

No gatherings of

≥ 9 individuals

No gatherings of

≥ 5 individuals

No gatherings of

≥ 3 individuals

(permission of 2 individuals after 6 PM)

Events

Prior notification to local governments for events with ≥ 500 people

Prohibition of events

with ≥ 100 people

Prohibition of events

with ≥ 50 people

Prohibition of any events

Nightlife entertainment facilities

No limitation

Restriction of operation after midnight

Restriction of operation

after 10 PM

Prohibition of gathering

Restaurants and cafes

No limitation

Permission of only packaging and delivery after midnight

Permission of only packaging and delivery after 10 PM

Permission of only packaging and delivery after 10 PM

November 2020

~ June 2021

1

1.5

2

2.5

3

Definition and major call to actions

Life quarantine

Local epidemic stage

National epidemic stage

< 100 casesa in Seoul

Metropolitan area

≥ 100 casesa in Seoul

Metropolitan area

≥ doubled over 1.5 stage or ≥ 300 casesa nationwide

≥ 400~500 casesa nationwide or rapid increase in patients

≥ 800~1000 casesa nationwide or rapid increase in patients

l Compliance with quarantine rules for COVID-19 prevention while maintain daily life and socioeconomic activity

l Start of local epidemic

l Through quarantine in dangerous areas

l Rapid propagation of local epidemic

l Restraint of unnecessary going out and gatherings in the risk area

l Spread nationwide epidemic

l Preferably stay at home and refrain from going out, meetings and using multi-use facilities as much as possible

l Nationwide pandemic

l As a rule, stay at home and minimize contact with others

Nightlife entertainment facilities

Limited to 1 person

per 4 m2

Adding no movement between seats

Prohibition of gathering

Restaurants and cafes

1 meter distance between tables in places over 150 m2

Expanding the distance in places

over 50 m2

Adding only permission of packaging and delivery after 9 PM

Adding the limitation to 1 person per 8 m2

Mandatory to wear a mask

(Penalties for violations)

All indoor facilities including public transportation

Adding outdoor sports arena

All indoor +

outdoor with high-risk activities

All indoor +

outdoors where it is difficult to keep a distance of 2 meters

Meeting and events

Consultation with local governments for gatherings of ≥ 500 +

Mandatory observance of basic quarantine rules

No gatherings of

≥ 100 people at special events such as festivals

No gatherings of

≥ 100 people

No gatherings of

≥ 50 people

No gatherings of

≥ 10 people

Working at workplace

Recommendation to activate telecommuting/homeworking

≥ 2/3 recommended to work from home

Mandatory telecommuting except for essential personnel

Going to school

1/3 of the density, in principle

Observe 2/3 of the density

1/3 of the density, in principle

Observe 1/3 of the density

Full remote learning

aAverage PCR-confirmed patients per week

References:

  1. Enhanced Social Distancing Campaign, 2020.4.4

: http://ncov.mohw.go.kr/searchBoardView.do?brdId=3&brdGubun=32&dataGubun=321&ncvContSeq=1497

  1. Central Quarantine and Countermeasure Headquarters. The COVID-19 Guidelines of the Republic of Korea for distancing in daily life

   : http://ncov.mohw.go.kr/en/guidelineView.do?brdId=18&brdGubun=181&dataGubun=&ncvContSeq=2763&contSeq=2763&board_id=&gubun=#

  1. Central Quarantine and Countermeasure Headquarters. Overview of Social Distancing System

: http://ncov.mohw.go.kr/en/socdisBoardView.do?brdId=19&brdGubun=191&dataGubun=191&ncvContSeq=&contSeq=&board_id=&gubun=

  1. Results

(1) Table 1: After COVID-19, PCP-suspected inpatients decrease among immunocompromised diseases, SOT recipients and Hematologic malignancies and have no changes among HSCT recipients and HIV-1 infection. In which group was the frequency of PCP-suspected inpatients increased? In non-immunosuppressed patients? If it is the case, the probability of confirm a PCP is lower and the rate of PCP-suspected and confirmed inpatients should be also lower. This point should consider in the discussion.

→ According to this point, we further retrieved the clinical information for underlying diseases in PCP-suspected inpatients and included the “solid cancer”, “chronic lung disease”, and “high-dose and lung-term corticosteroid therapy”, which are all risk factors for PCP development. The frequency of PCP-suspected cases among patients undergoing high-dose and long-term corticosteroid therapy (≥20 mg/day of prednisone or equivalent when administered for ≥14 consecutive days), which is another immunocompromised group at higher risk for PCP, was not different between pre- and post-COVID-19 period (1545 [10.9%] vs. 679 [11.5%], P = 0.512). However, the frequency of solid cancers in PCP-suspected inpatients was significantly higher in the post-COVID-19 period compared to pre-COVID-19 (1651 [28.1] vs. 3561 [25.1], P < 0.001). The frequency of PCP-suspected cases among patients with chronic lung disease was not different between pre- and post-COVID-19 period (243 [1.7%] vs. 112 [1.9%], P = 0.305). When we included the solid cancers, chronic lung diseases, and high-dose/long-term corticosteroid therapy in the immunocompromised conditions, the frequency of PCP-suspected inpatients with the immunocompromised conditions (SOT/HSCT recipients, solid cancers, hematologic malignancies, HIV-1 infection, chronic lung diseases, and high-dose/long-term corticosteroid therapy) was similar between pre- and post-COVID-19 period (8531 [60.1%] vs. 3508 [59.6%], P = 0.872).

→ We decided that we could not determine the further detailed immunocompromised or immunosuppressive co-morbid diseases or conditions, because we could not clearly categorize other severe immunocompromised status at high risk for PCP except for those included in Table 1. Besides, the immunocompromised conditions in the revised table 1 was not different between pre- and post-COVID-19 period. For these reasons, we changed the table 1, 2, 3, 4 as well as all figures in the whole revised manuscript including further data for solid cancers, chronic lung diseases and high-dose, long-term corticosteroid therapy (Please see the revised tables and figures).

→ We inserted the definition for high-dose and long-term corticosteroid therapy and chronic lung diseases in Materials and Methods section.

  • [ Materials and Methods - 2.1. Study design and data collection: Line - ]: Data regarding age, sex, index date, history of ICU admission, all-cause of death during hospitalization, and immunocompromised status were collected from the RDBMS. We stated the immunocompromised conditions as higher risk morbidities for PCP for solid organ (SOT) or hematopoietic stem cell transplantation (HSCT), solid cancers, hematologic malignancies, human immunodeficiency virus (HIV)-1 infection, chronic lung diseases (chronic obstructive pulmonary disease, interstitial lung disease, and idiopathic pulmonary fibrosis), and corticosteroid therapy in this study [18,51]. The high-dose and long-term corticosteroid therapy was defined as ≥20 mg/day of prednisone or equivalent when administered for ≥14 consecutive days [52].

  • [New references]:
  1. 51. vino, L.J.; Naylor, S.M.; Roecker, A.M. Pneumocystis jirovecii pneumonia in the non-hiv-infected population. Ann Pharmacother 2016, 50, 673-679.

52.National Center for Immunization and Respiratory Diseases. General recommendations on immunization --- recommendations of the advisory committee on immunization practices (acip). MMWR Recomm Rep 2011, 60, 1-64.

→ After the addition of the data for solid cancer, chronic lung diseases and corticosteroid therapy, the final conclusion as well as the results for total PCP-suspected inpatients in ARIMA and Bayesian structural time-series model were not altered.

→ But, the data of time-series analyses in patients with solid cancer and corticosteroid therapy were changed. We revised the main text, Table 2~4, figure 3D, and supplementary figure 2D.

  • [ Results- 3.1. Clinical information of total PCP-suspected and -confirmed inpatients: Line - ]: The mean age and percentage of male patients in the total PCP-suspected patients (N=20,073) were 65 years and 64.7%, respectively; 29.1%, 24.5%, and 60.0% of total PCP-suspected patients had a history of ICU care, all-cause mortality during admission at PCP diagnosis, and immunocompromised status, respectively. PCP-suspected patients in the post-COVID-19 period (N=5,881) had a significantly higher age (65.7 ± 14.5 vs. 64.0 ± 14.6, P<0.001) and lower frequencies of ICU care (27.8% vs. 29.7%, P=0.006) than PCP-suspected patients in the pre-COVID-19 period (N=14,192). The frequency of immunocompromised status was similar between two periods (60.1% and 59.5%, P=0.872). The PCP-suspected inpatients in the post-COVID-19 period had the significantly higher rate of solid cancers, but lower rate of SOT and hematologic malignancies compared to those in the pre-COVID-19 period (all P<0.001) (Table 1).

  • [ Results- 3.1. Clinical information of total PCP-suspected and -confirmed inpatients: Line - ]: The ICU care at admission (P=0.001) in PCP-confirmed patients were significantly higher than that in suspected patients without PCP in the pre-COVID-19 period. This characteristic was not different between the two patient groups in the post-COVID-19 (P=0.487). The percentage of patients with immunocompromised conditions was similar between two groups in the pre- and post-COVID-19 periods (Table 2).

  • [ Results- 3.2. Trend analysis and ARIMA model to compare the observed and forecasted PCP rates post-COVID-19: Line - ]: The ARIMA model did not reveal a significant difference between the observed and forecasted PCP rates in total PCP-suspected inpatients in the post-COVID-19 period (-2.4% of residual estimate, P=0.102). In subgroup analyses, the observed PCP rates in patients with immunocompromised conditions (-2.5% of residual estimate and 0.8% of standard error, P=0.004), particularly in SOT recipients (-7.8% and 3.1%, P=0.016), solid cancers (-2.4% and 0.9%, P=0.008), steroid therapy (-5.2% and 0.8%, P<0.001), and in the younger group (<65 years, -3.5% and 1.5%, P=0.027), were significantly lower than the forecasted PCP rates in the post-COVID-19 (Table 3). Regardless of the significant decline in estimated residuals in the ARIMA model, the plots and regression lines of the observed PCP rates in the immunocompromised or younger groups did not show the distinctly downward patterns during the post-COVID-19 period, with the generally lowest rates in November and December 2020. In addition, the observed PCP rates were similar between the latest time point (June 2021) in the post-COVID-19 period and the latest month (December 2019) in the pre-COVID-19 in the total patients and most subgroups (Supplementary Figure 1). The observed and forecasted PCP rates in patients without immunocompromised conditions and critically ill patients receiving ICU care did not differ in the post-COVID-19 (Table 3).

  • [ Results- Bayesian structural time-series model to compare observed and predicted PCP rates in the post-COVID-19: Line - ]: The overall observed PCP rate (13.0%) in total PCP-suspected inpatients in the post-COVID-19 was lower than that (16.0%) in the pre-COVID-19. However, the counterfactually predicted PCP rates (14.0%, 95% CI: 11.9-16.1%) in the post-COVID-19 by the Bayesian structural time-series model were not significantly different from the observed PCP rates in the post-COVID-19 in total PCP-suspected patients (absolute effect, -1%, 95% CI: [‒3 ~ 1%], and relative effect, -7% [-22 ~ 8%], P=0.183) (Table 4 and Figure 3A). In subgroup analyses, the patients with immunocompromised conditions (‒1% [‒3 ~ ‒0.2%] and -9% [‒17 ~ ‒1%], P=0.012), particularly in solid cancers (‒3% [‒6 ~ ‒1%] and ‒21% [‒34 ~ ‒6%], P=0.002) and steroid use (‒5% [‒6 ~ ‒4%] and ‒37% [‒47 ~ ‒27%], P=0.001) had significantly higher observed PCP rates (17%) than the predicted rates (5%) in the post-COVID-19 (Table 4 and Figure 3D). The cumulative residuals of PCP rates in patients with immunocompromised conditions including solid cancers and steroid use steadily decreased with statistical significance in the post-COVID-19 period (Figure 3D).

→ We mentioned this altered changes in the “Discussion” section.

  • [ Discussion: Line - ]: Among several characteristics of PCP-suspected inpatients, solid cancer and corticosteroid therapy had a significant alteration in PCP rates in the post-COVID-19 in two time-series models. However, we could not affirm the continuous and dramatic reduction of monthly observed PCP rates even in the patients with solid cancer and steroid therapy (Figure 4D and Supplementary Figure 1). Long-term follow-up studies will be needed in the specific immunocompromised patients with high risk for PCP.

(2) Table 3: Relative average effect of 222.2 (49.0~373.0) in HSCT patients is right? Please, confirm it.

→ We rechecked the original data and performed the statistical analysis (Bayesian structural time-series model) again. Afterward, we confirmed the original data is correct. We presumed that this large value of relative average effect in HSCT recipients might be caused by absence of PCP-confirmed cases in several months as well as relative small number of total inpatients (N = 249) in this subgroup.

→ As with all data (total N=20,073), the monthly number of PCP-confirmed case must be secured to some extent at least 20 per group to achieve stable and appropriate estimation, whereas it may have been overestimated (overfitted) because a sufficient number in HSCT subgroup was not secured.

[Supplementary Figure 1. Plots and liner regression lines of the observed PCP rates in PCP-suspected- inpatients in the pre- and post-COVID-19 periods. D) Immunocompromised diseases – HSCT recipient subgroups]

[SOT recipient subgroups]

→ We previously mentioned this point in the original manuscript (“Discussion” section)

: Among several characteristics of PCP-suspected inpatients, SOT and HSCT recipients had a significant alteration in PCP rates in the post-COVID-19 in two time-series models. However, these findings would most likely be the unreliable results arising from random fluctuations unrelated to the strict NPIs and COVID-19 pandemic because the PCP in these groups did not occur in several months during the pre-COVID-19 period (Supplementary Figure 1).

→ However, we decided to exclude three subgroups of “HSCT recipients”, “HIV-1 infection” and “Chronic lung disease” in two time-series analyses to verify the quality of statistical analysis.

  • [ Results- 3.2. Trend analysis and ARIMA model to compare the observed and forecasted PCP rates post-COVID-19: Line - ]: As the observed PCP-confirmed cases and expected PCP rates did not exist for several months in the pre- and post-COVID-19 periods, we did not perform the time-series analysis for HSCT recipients, chronic lung disease, and HIV-1-infected individuals in the ARIMA and Bayesian structural time-series model.

→ And, we inserted this point as our limitation in the “Discussion” section.

  • [ Discussion: Line - ]: Forth, we could not analyze the specific immunocompromised subgroups, because the PCP did not occur in several months during the pre-COVID-19 period and time-series analyses might generate the unreliable results arising from random fluctuations.

  1. Discussion

(1) Some articles have shown the risk of Pneumocystis infection among household. 

  • Morilla R, et al. Pneumocystis jiroveciiamong patients with cystic fibrosis and their household members. Med Mycol. 2021 Mar 10:myab010. doi: 10.1093/mmy/myab010.
  • Rivero L, et al. Pneumocystis jiroveciitransmission from immunocompetent carriers to infant.  Emerg Infect Dis. 2008 ;14: 1116-8. doi: 10.3201/eid1407.071431.

Intra-family transmission of P. jirovecii could be considered in the discussion as one possible cause to explain the no mitigation of PCP incidence with the measures against the COVID-19 pandemic because face masks and social distancing do not apply in family environment.

→ We do appreciate for this important comment. Because we fully agree with the reviewer’s suggestion, we newly added the below sentence about intra-family transmission with the suggested references in the revised manuscript.

  • [ Discussion: Line - ]: In addition, the intra-family transmission of P. jirovecii could be as one possible cause to explain no mitigation of PCP incidence with the stricter NPIs against the COVID-19 pandemic, because face masks and social distancing do not apply to family in the house and some reports have shown the risk of P. jirovecii exposure among household members [69,70].

  • [New references]
  1. Rivero, L.; de la Horra, C.; Montes-Cano, M.A.; Rodríguez-Herrera, A.; Respaldiza, N.; Friaza, V.; Morilla, R.; Gutiérrez, S.; Varela, J.M.; Medrano, F.J. et al. Pneumocystis jirovecii transmission from immunocompetent carriers to infant. Emerg Infect Dis 2008, 14, 1116-1118.
  2. 70. Morilla, R.; Medrano, F.J.; Calzada, A.; Quintana, E.; Campano, E.; Friaza, V.; Calderón, E.J.; de la Horra, C. Pneumocystis jirovecii among patients with cystic fibrosis and their household members. Med Mycol 2021,doi:10.1093/mmy/myab010.

  1. Conclusion: (line 375-377) ‘Taken together our time-series analyses suggest that P. jiroveciimay not be mainly acquired by airborne person-to-person transmission’. I think that based exclusively on the results of this study is unwise to assume this affirmation.

→ We fully consent to the reviewer’s comment. This problem is also mentioned by the Reviewer #2

(I suggest changing the title which does not illustrate the content of the manuscript, changing the introduction by moving some paragraphs from the discussion, discussing better and frankly the many biases of the study, and changing the conclusion to saying that the present study cannot support any mode of transmission. 7. Conclusion: Q: To be changed.)

→ Because we total agree with this issue, we deleted the sentence of “Taken together, our time-series analyses suggest that P. jirovecii may not be mainly acquired by airborne person-to-person transmission” in the revised manuscript. And, we inserted the below sentence in the “Conclusion section”

  • Further large studies are needed to uncover the detailed epidemiologic evidences and pathogenic mechanisms of PCP development.

Reviewer 2 Report

This manuscript presents the epidemiology of PCP in 4 Korean hospitals before and after mask wearing due to COVID-19. Since the number of PCP did not change, which might have been expected due to the known airborne transmission of Pneumocystis jirovecii, the authors conclude that their results make airborne transmission of Pneumocystis jirovecii between people unlikely. Discussions are endless about the occurrence of PCP, whether it is the reactivation of latent contamination or new airborne contamination, but the present study does not support one hypothesis over the other.

The authors' initial hypothesis is that wearing a mask should have stopped the transmission of P. jirovecii and therefore reduced the number of PCP. However, in PCP, transmission and disease are not directly related if patients at risk for PCP are not hospitalized and tested as in the first period. Indeed, due to the profound evolution of the hospitalization policy, the patients at risk of PCP were probably not hospitalized at the same rate between the two periods (for example, the solid organ transplant program ceased, non-urgent cancer complications were not hospitalized, or hospitalization postponed…). There are so many variables that have changed between pre- and post-COVID-19 that it seems unlikely that the sampling and the clinical decision to treat or not are comparable. Therefore, I don't see how statistical algorithms can calculate a projection useful for comparison when so many variables have changed.

On the other hand, the prevalence of PCP depends on the clinical decision whether or not to test for P. jirovecii in COVID-19 patients. A systematic search can find up to 9.3% (10/108 patients) of ICU patients with COVID-19 (Alanio A et al, The presence of Pneumocystis jirovecii in critically ill patients with COVID-19, Journal of Infection 82 (2021) p 114) although not all PCR positive patients received cotrimoxazole. Therefore, disease and transmission could not be simply linked as suggested above. Instead of trying to define PCP after clinicians' decisions or imaging (no one knows about PCP imaging in COVID-19 patients), which is a huge additional bias, I wonder if measuring the number of positive PCR tests would be more objective to see the environmental circulation of the fungus. Indeed, a P. jirovecii PCR positive patient without supportive imaging and not treated by the clinician does not mean that the mask is useless to prevent transmission (and quite the contrary if no outbreak occurred). Instead of looking for supportive imaging, I would have paid more attention to the PCR results, central to the diagnosis, and analyzed the quantitative PCR results. As a control, I would have tested an airborne virus to see the effectiveness of wearing the mask. If virus transmission declines and positive PCR results do not drop, it would have been more suggestive of the pointlessness of wearing the mask for PCP. I do recognize, however, that this is a tremendous amount of extra work.

I suggest changing the title which does not illustrate the content of the manuscript, changing the introduction by moving some paragraphs from the discussion, discussing better and frankly the many biases of the study, and changing the conclusion tosaying that the present study cannot support any mode of transmission.If the raw PCR results (number of positive tests / number of tests performed without clinical interpretation, and if possible, quantitative results) could be presented and discussed, this could also be informative for the circulation of the fungus.

Specific comments

Title: Change in Incidence Rate of Pneumocystis jirovecii Pneumonia after the COVID-19 Pandemic: Multicenter Time-series Analyses

Q: Since no change was observed, say “no change” in the title.

Based on the issue of whether Pneumocystis jirovecii could be transmitted by airborne or acquired from the environment

Q airborne and acquired

The number of COVID-19 patients confirmed by real-time reverse transcription polymerase chain

….

such as restaurants, at night. 51

Q: This paragraph is interesting but not directly linked to the present study and could be shortened.

We thoroughly reviewed the chest radiologic findings of patients who had received diagnostic tests for P. jirovecii, assigned as PCP-suspected patients, to verify PCP diagnoses and exclude asymptomatic P. jirovecii colonization or unnecessary PCP tests. The affirmation of chest radiologic findings was performed using a text search of the radiologists’ readings with the keywords 100 ‚”pneumonia”, “consolidation”, and “infiltration” in the RDBMS [33,34]. 101

Q: In doing so, the authors do not control for the patients who had imaging compatible for PCP and for whom no P. jirovecii test was requested.

After excluding 734 outpatients and 27 inpatients with unnecessary PCP tests,

Q: That could be interesting in term of circulation of the fungus to discuss the results when the test was “unnecessary”.

2.3. P. jirovecii PCR

The qualitative real-time PCR tests for P. jirovecii

Q: If the PCR format is real-time PCR, it would mean that the result can be expressed in quantitative cycle, which is much more informative for the circulation of the fungus than a yes or no result. Authors should attempt to quantify their PCR results to analyze transmission. Indeed, even a low fungal load can participate in the transmission (see for instance doi: 10.3389/fmicb.2017.00700).

Discussion

Lines 310-324 The various forms of P. jirovecii during the life cycle in humans are approximately

of airborne HCW-to-patient (and vice versa) or person-to-person transmission of P. jirovecii in intra-hospital or community settings.

Q: This long paragraph should be in the introduction, as a rationale for the study. This paragraph does not discuss the results.

Line 330- 349

Q: This long paragraph actually discusses the likelihood that "colonized" patients may have played a role.As it is neither explored nor controlled in the present study, the authors should present this paragraph as a limitation of their study.The last part of the paragraph on serology should be in the introduction.

Line 350: This first attempt to analyze the potential influence of stricter NPIs during the COVID-19 pandemic on PCP rate has some strengths: (1) complete exclusion of asymptomatic P. jirovecii colonization in PCR test,

Q: As said above, I think it is rather a weakness than a strength when the issue is airborne transmission.

Line 358: However, our study has some limitations. First, a major hurdle was the convoluted selection of the exposed or target population to obtain an accurate incidence rate of PCP.

Q: The “convoluted” aspect of the selection should be discussed earlier since it is a major limitation for the study.

Conclusion

Q: To be changed.

Author Response

General comments:

This manuscript presents the epidemiology of PCP in 4 Korean hospitals before and after mask wearing due to COVID-19. Since the number of PCP did not change, which might have been expected due to the known airborne transmission of Pneumocystis jirovecii, the authors conclude that their results make airborne transmission of Pneumocystis jirovecii between people unlikely. Discussions are endless about the occurrence of PCP, whether it is the reactivation of latent contamination or new airborne contamination, but the present study does not support one hypothesis over the other.

→ We fully understand your concerns. Basically, our research started with the observation for seasonal influenza and other community-acquired respiratory viruses (CA-RVs), transmitted by air (aerosol and droplet), during COVID-19 pandemic. As you know and we have fully described these phenomena in the previous paper, the occurrence and incidence rates of influenza and CA-RVs has been dramatically reduced after COVID-19 pandemic in several countries. The strict and enhanced NPIs including mandatory mask wearing, social distancing, and other intensive control measures with unprecedented great compliance (not limited to wearing a mask), to prevent the public and intra-hospital transmission (nosocomial outbreak) against SARS-CoV2, could sufficiently affect the spread of other airborne transmitted pathogens.

→ Even though our research was not directly designed (particularly using molecular surveillance study), to evaluate whether the acquisition of Pneumocystis jirovecii (possibly transmissible fungi by air from many previous studies) and/or development of symptomatic PCP disease may be arisen from reactivation of latently colonized asymptomatic fungus or new transmission by air or environment or other sources, our time-series analyses considering real clinical setting in the current medical situation, which may be difficult to re-experience, may be helpful to understand the epidemiology of P. jirovecii and PCP. If P. jirovecii spreads and transmitted mainly through the air, PCP rates could be significantly decreases by these strict infection control measures, like as influenza virus and CA-RVs.

→ We think that our study may be fundamentally similar with previously reported researches about the change of proportion of influenza virus (rate of symptomatic PCP disease) among ILI (influenza-like illness) (PCP-suspected inpatients in our study) between two periods (before and after COVID-19 pandemic). (Please see our answers with some references for the reviewer’s below comments.)

With all these considerations in mind, we ask for a deep understanding of the originality and clinical significance (usefulness and suggestion for further study in the field of P. jirovecii epidemiology, currently not fully understood) of our research.

The authors' initial hypothesis is that wearing a mask should have stopped the transmission of P. jirovecii and therefore reduced the number of PCP. However, in PCP, transmission and disease are not directly related if patients at risk for PCP are not hospitalized and tested as in the first period. Indeed, due to the profound evolution of the hospitalization policy, the patients at risk of PCP were probably not hospitalized at the same rate between the two periods (for example, the solid organ transplant program ceased, non-urgent cancer complications were not hospitalized, or hospitalization postponed…). There are so many variables that have changed between pre- and post-COVID-19 that it seems unlikely that the sampling and the clinical decision to treat or not are comparable. Therefore, I don't see how statistical algorithms can calculate a projection useful for comparison when so many variables have changed.

→ We also fully agree with the reviewer’s valuable comments and opinions.

→ First of all, we checked the number of outpatients who were received PCP tests (PCR and cytology) at outpatient clinic during total study period (after exclusion of the repeated tests) (PCP-suspected outpatients).

  • Total period (2015.1~2021.6): 701 outpatients (5% compared to inpatients [N = 20,073])
  • Before COVID-19: 474 outpatients (3% compared to inpatients [N = 14,192])
  • After COVID-19: 227 outpatients (9% compared to inpatients [N = 5,881])

 (1) The ratios of inpatients and outpatients were similar between pre- and post-COVID-19 period.

 (2) The percentages of outpatients compared to inpatients were quite low in the total duration, as well as pre- and post-COVID-19 periods.

→ The numbers of PCP-confirmed outpatients were as follows:

  • Total period (2015.1~2021.6): 67 outpatients (3% compared to inpatients [N = 2,922])
  • Before COVID-19: 50 outpatients (3% compared to inpatients [N = 2,163])
  • After COVID-19: 17 outpatients (2% compared to inpatients [N = 759])

(1) The ratios of PCP-confirmed inpatients and outpatients were similar between pre- and post-COVID-19 period.

 (2) The percentages of PCP-confirmed outpatients compared to inpatients were quite low in the total duration, pre- and post-COVID-19.

→ Furthermore, 41 PCP-confirmed outpatients were hospitalized after first PCP diagnosis at outpatient clinics and already included in our original data. Therefore, the pure number of PCP-confirmed and PCP-suspected outpatients (not hospitalized) was only 26 and 660 during total study period (6.5 years – 78 months), respectively.

→ This real data showed that the patients at risk of PCP (PCP-suspected patients) were hospitalized at the same rate between the pre- and post-COVID period. The majority (almost all – 99.1% [2,896 of 2,922]) of PCP-confirmed patients were hospitalized.

→ We fully agree with this point: “The development of PCP and transmission of P.jirovecii are not directly related to hospitalization, and may be associated with many variables.” However, we confirmed that almost all PCP-confirmed and –suspected patients were hospitalized through further data mining. Therefore, we decided that addition of outpatient’s data does not make much sense in the revised manuscript, because the new very small data (26 and 660 cases during 78 months) did not actually affect the final result and our conclusion in time-serises models based on monthly observed data. We ask for your deep understanding of our well-thought-out judgment in making a decision based on these details.

→ To our best knowledge, the traditional statistical analyses or algorithms could not control so many potential contributing variables or confounders between two periods (before and after intervention). To overcome this difficulty, new time-series models (Bayesian structural time-series model), introduced by Google team, forecast the predicted values after intervention (stricter NPI after COVID-19 pandemic in this study) using the real observed data (ground truth) before intervention through knotty mathematical and statistical algorithms. Afterward, we calculate the residuals between the predicted and observed data in the post-intervention period, and decide if the alteration of interest after intervention is significant and meaningful. The time-series models can forecast the predicted values after intervention through their own algorithms without any consideration of variables that have changed between pre- and post-COVID-19.

(Please refer this original article for the detailed math formula- (1) The Annals of Applied Statistics, 2015, Vol. 9, No. 1, 247–274, https://doi.org/10.1214/14-AOAS788, INFERRING CAUSAL IMPACT USING BAYESIAN STRUCTURALTIME-SERIES MODELS, Google, Inc. (2) The effect of bariatric surgery on health care costs: A synthetic control approach using Bayesian structural time series. Health Economics. 2019;28:1293–1307. DOI: 10.1002/hec.3941.2.2 Bayesian structural time series].

→ The time-series analyses including Bayesian structural time-series model or ARIMA model have been widely used in the medical epidemiology studies and the evaluation of public health interventions. We strongly believe that our time-series statistical methods (ARIMA model and Bayesian structural time-series model) enough provide a sufficient solution for the reviewer’s concerns.

→ The reporting or sample surveillance system also have the limitation of underreporting or a lapse of surveillance. However, we included the all PCP-suspected cases without any sampling which could have any selection biases.

→ Please refer the below several references and figures for change of incidence rates before and after intervention [including COVID-19 pandemic] using ARIMA and Bayesian structural time-series model.

(1) Impact of Nonpharmaceutical Interventions on the Incidence of Respiratory Infections During the Coronavirus Disease 2019 (COVID-19) Outbreak in Korea: A Nationwide Surveillance Study: Clinical Infectious Diseases 2021;72(7):e184–91, DOI: 10.1093/cid/ciaa1682 – Compare the incidence rates of airborne-transmitted pathogens between pre- and post-COVID pandemic period without considering any changed variables using ARIMA models

(2) Reduction in Kawasaki Disease After Nonpharmaceutical Interventions in the COVID-19 Era. Circulation. 2021;143:2508–2510. DOI: 10.1161. Compare the incidence rates of Kawasaki disease between pre- and post-COVID pandemic (NPI) period without considering any changed variables using ARIMA models

(3) The effect of bariatric surgery on health care costs: A synthetic control approach using Bayesian structural time series. Health Economics. 2019;28:1293–1307. DOI: 10.1002/hec.3941. Compare the expenditures before and after intervention (surgery) without considering any changed variables using the Bayesian structural time-series models

→ We previously addressed this point in detail with several references in the original manuscript.

  • [ Materials and Methods: 2.5. Statistical analysis] - Additionally, we employed the Bayesian structural state-space model for time-series, which was proposed by Google Inc. in 2015 and has been implemented in public health research, using SAS version 9.4 and R language (version 4.1.0) (http://www.r-project.org) with the CausalImpact package (http://google.github.io/CausalImpact/) [60-62]. This Bayesian model determines the causal impact of a planned intervention by acquiring a counterfactual prediction in an artificial control of what would have taken place had this intervention not occurred [60,61]. The intervention in this study indicated that stringent nationwide NPIs arose due to the COVID-19 pandemic. A counter fact, which implied that PCP rates would have developed if the COVID-19 pandemic did not occur, was obtained from the true (observed) PCP rates during the pre- and post-COVID-19 periods. The causal impact of enforced national and hospital policies on PCP rates was estimated by calculating the pointwise (each month) and cumulative (the whole post-COVID-19 period) residuals, which was the distinction between the overall observed and counterfactual (predicted) PCP rates in the post-COVID-19. The average absolute (observed–predicted) and relative ([observed-predicted]/predicted×100) casual effects caused by the intervention (COVID-19 pandemic) were expressed as percentages and 95% counterfactual prediction credible intervals (CIs) [63].

On the other hand, the prevalence of PCP depends on the clinical decision whether or not to test for P. jirovecii in COVID-19 patients. A systematic search can find up to 9.3% (10/108 patients) of ICU patients with COVID-19 (Alanio A et al: The presence of Pneumocystis jirovecii in critically ill patients with COVID-19, Journal of Infection 82 (2021) p 114) although not all PCR positive patients received cotrimoxazole. Therefore, disease and transmission could not be simply linked as suggested above. Instead of trying to define PCP after clinicians' decisions or imaging (no one knows about PCP imaging in COVID-19 patients), which is a huge additional bias, I wonder if measuring the number of positive PCR tests would be more objective to see the environmental circulation of the fungus.

→ We think that that’s exactly right (“no one knows about PCP imaging in COVID-19 patients”). We first ask for a deep understanding of the following: (1) We did not analysis PCP rates (results of PCP PCR test) in COVID-19-infected patients, (2) We analyzed the change of PCP rates between pre- and post-COVID-19 periods, (3) In our data, we did not have co-infected patients by SARS-CoV-2 and P. jirovecii, (4) Our study has no relevance to the COVID-19 patients. We analyzed the PCP rates among all PCP-suspected inpatients between pre- and post-COVID-19 pandemic, (5) COVID-19 in our research just have the meaning as the intervention (strict NPIs including wearing of mask and infection control measures) against SARS-CoV2.

→ As noted in comments (“if measuring the number of positive PCR tests would be more objective to see the environmental circulation of the fungus”), we measured the number of positive PCP PCR tests (namely, PCP-confirmed patients) in the original manuscript.

Indeed, a P. jirovecii PCR positive patient without supportive imaging and not treated by the clinician does not mean that the mask is useless to prevent transmission (and quite the contrary if no outbreak occurred). Instead of looking for supportive imaging, I would have paid more attention to the PCR results, central to the diagnosis, and analyzed the quantitative PCR results. As a control, I would have tested an airborne virus to see the effectiveness of wearing the mask. If virus transmission declines and positive PCR results do not drop, it would have been more suggestive of the pointlessness of wearing the mask for PCP. I do recognize, however, that this is a tremendous amount of extra work.

→ We completely agree with this point. We think that the following sentence as noted in comment is exactly right (“P. jirovecii PCR positive patient without supportive imaging and not treated by the clinician does not mean that the mask is useless to prevent transmission”). We analyzed the ratio of P. jirovecii PCR positive inpatients (supported by imagining and treated by the clinicians, PCP-confirmed inpatients) to all inpatients who were received P. jirovecii tests including PCR (PCP-suspected inpatients) same as the reviewer’s comment.

→ We think what does not take into account the image result (suspicion of PCP or not) would be same that influenza tests in surveillance system are implemented in all or any visitors at outpatient clinic regardless of symptoms (influenza like illness). We ask for your deep understanding of our thoughts.

→ When we searched all P. jirovecii PCR test in both inpatients and outpatients during all study period (between 2015.1 and 2021.6), we just found only three P. jirovecii PCR positive patient among total 27 cases (named as unnecessary P. jirovecii PCR test) without supportive PCP imaging and not treated by the clinician. According to the below specific comment, we described this point in the revised manuscript.

→ In addition, we carefully checked whether the rates of PCP cases (positive PCP PCR tests) could be underestimated or overestimated by the change of total number of PCP tests between two periods.

As noted in comments, the study for quantitative PCP PCR in air samples and the test for airborne viruses as control to see the effectiveness of wearing the mask will be ideal design. We ask for your deep understanding for importance and usefulness of our study design (comparing to influenza and other CV-RVs in same real medical situation, with reference to previous well-designed and well-done several manuscripts). We ask for your deep understanding that this study was not aimed at the pointlessness or usefulness of mask for P. jirovecii.

I suggest changing the title which does not illustrate the content of the manuscript, changing the introduction by moving some paragraphs from the discussion, discussing better and frankly the many biases of the study, and changing the conclusion to saying that the present study cannot support any mode of transmission. If the raw PCR results (number of positive tests/number of tests performed without clinical interpretation, and if possible, quantitative results) could be presented and discussed, this could also be informative for the circulation of the fungus.

→ We also fully agree with the reviewer’s suggestion and revised the manuscript.

(1) Title: No Change of Pneumocystis jirovecii Pneumonia after the COVID-19 Pandemic: Multicenter Time-series Analyses

(2) Introduction: According to the reviewer’s recommendation at specific comments, we moved the paragraph of “Discussion” into the “Introduction” section. Please refer to our detailed responses and changes for specific comments.

(3) Discussion: According to the reviewer #1’s comments and the below specific comments, we did our best to improve the description in the “Discussion” section.

(4) Conclusion:

→ Because we total agree with this issue, we deleted the sentence of “Taken together, our time-series analyses suggest that P. jirovecii may not be mainly acquired by airborne person-to-person transmission” in the revised manuscript. And, we inserted the below sentence in the “Conclusion section”

  • Further large studies are needed to uncover the detailed epidemiologic evidences and pathogenic mechanisms of PCP development.

(5) Raw PCR results: We added the description for number of positive tests/number of tests performed without clinical interpretation (named as unnecessary P. jirovecii PCR test) in the “Discussion” section (Please see our answers in the specific comments.)

Specific comments

  1. Title: Change in Incidence Rate of Pneumocystis jirovecii Pneumonia after the COVID-19 Pandemic: Multicenter Time-series Analyses

Q: Since no change was observed, say “no change” in the title.

→ According to the reviewers’ comments for title (The reviewer #1 recommended that it is better: ‘Pneumocystis jirovecii Pneumonia after the COVID-19 Pandemic: Multicenter Time-series Analyses, because there is no change in incidence rate’.) we revised the title as the below sentence

  • Title: No Change of Pneumocystis jiroveciiPneumonia after the COVID-19 Pandemic: Multicenter Time-series Analyses

  1. 2. Abstract: Based on the issue of whether Pneumocystis jirovecii could be transmitted by airborne or acquired from the environment

Q: airborne and acquired

→ According to the comment, we revised it.

  1. 3. Introduction: The number of COVID-19 patients confirmed by real-time reverse transcription polymerase chain…. such as restaurants, at night.

Q: This paragraph is interesting but not directly linked to the present study and could be shortened.

→ We fully agree with the reviewer’s point. But, the reviewer #1 recommended that we should include “information about when and what non-pharmacological interventions against COVID-19 pandemic imposed by Korean government” in the “Materials and Methods” section. Taking these comments into account, we moved the whole paragraph into the “Materials and Methods” section and included the detailed description about NPIs in the “Materials and Methods” section.

  • [ Materials and Methods: 2.2. Non-pharmacological interventions against COVID-19 pandemic imposed by the Korean government: Line - ] - After the first large outbreak in relation to religious gatherings in Dageu/Gyeongbuk province at the end of February 2020, the Korean government had executed the emergent enforced social distancing with measures to restrict the operation of multi-use facilities at risk of mass transmission, prohibition of gathering, call to actions for all citizens or for citizens in the workplace, and recommendation for mask wearing until early May. Since then, while the relaxed social distancing in life has been implemented, the second outbreak in August resulted in the stricter enforcement of the enhanced social distancing again (2 or 3 out of 4 levels). The government changed the social distancing to the lowest level 1 from mid-October and reorganized the levels in the form of precision quarantine with five steps from early November 2020. The number of COVID-19 patients confirmed by real-time reverse transcription PCR in South Korea was 300-800 per day due to clustered infection from December 2020 to June 2021, which subsequently increased to 2,000 cases/day from July 2021 (WHO COVID-19 Dashboard, https://covid19.who.int/region/wpro/country/kr, Korean Central Disease Control Headquarters, http://ncov.mohw.go.kr/en/). Since the highest surge in new cases in early December 2020, the government has been maintaining the enhanced social distancing (2, 2.5, and 3 levels) to prohibit personal gatherings and businesses, such as restaurants, at night. until the end of June (See the details for social distancing levels in Supplementary Table 1). The strong recommendation of mask wearing, except mesh or valve type mask and cloths or scarfs, turned into compulsory fulfillment as a fine penalty for not wearing a mask in the multi-use facilities had been imposed from November 2020 [53-56].

  1. 4. Materials and Methods

(1) We thoroughly reviewed the chest radiologic findings of patients who had received diagnostic tests for P. jirovecii, assigned as PCP-suspected patients, to verify PCP diagnoses and exclude asymptomatic P. jirovecii colonization or unnecessary PCP tests. The affirmation of chest radiologic findings was performed using a text search of the radiologists’ readings with the keywords of “pneumonia”, “consolidation”, and “infiltration” in the RDBMS [33,34].

Q: In doing so, the authors do not control for the patients who had imaging compatible for PCP and for whom no P. jirovecii test was requested.

→ We fully consent to this comment. We think that the radiologic findings compatible for PCP would be the pneumonias caused by viruses, atypical (most commonly community-acquired) bacteria (Mycoplasma pneumonia, chlamydia trachomatis, and Legionella pneumophila) or rare opportunistic pathogens or specific microbes in certain severe immunocompromised patients (for instance, active tuberculosis in HIV-1-infected patients).

→ We think that it could be really hard to control for the patients who had imaging compatible for PCP and for whom no P. jirovecii test was requested in real clinical practice. We could not differentiate the PCP from other interstitial pneumonia with compatible radiologic findings (GGO pattern) by just chest X-ray and CT scan. It may be entirely at the discretion of the attending physicians. On the contrary to this, we will also need to consider the appropriate performance of diagnostic tests (diagnostic stewardship)

This control process might be allowed to time-series analyses to evaluate the changes of incidence rates before and after intervention periods in many epidemiologic research, because the researcher could not control the accurate diagnostic process in the (long-term) pre-intervention period, particularly in the research in relation to this emergent medical situation (COVID-19 pandemic). We ask for your deep understanding of these difficulties.

→ With such considerations, we included this point as the limitation of our study in the "Discussion” section.

  • [ Discussion – “paragraph presenting study limitation”: Line - ]: Third, the rates of PCP might be underestimated all in any particular period, because we could not consider the patients with interstitial pneumonia, but not any tests for P. jirovecii.

  • [ Discussion : Line - ]: Our study should be interpreted in consideration of the following points as probable biases: (1) difficult diagnostic process for P. jirovecii even in patients with symptomatic lung infiltration, (2) not control for the patients who had imaging compatible for PCP and for whom no P. jirovecii test was requested, (3) not surveillance tests in asymptomatic individuals, and (4) not all patients are hospitalized, even though our data had very low rate of PCP-confirmed (N=26) and –suspected (N=660) outpatients in the total duration of 78 months.

(2) After excluding 734 outpatients and 27 inpatients with unnecessary PCP tests,

Q: That could be interesting in term of circulation of the fungus to discuss the results when the test was “unnecessary”.

→ We fully agree with this valuable point. We directly reviewed the electric medical records for patients with “unnecessary PCP PCR tests” again. The 22 and 5 unnecessary PCP PCR tests were performed at pre- and post-COVID19 period, respectively. Three patients (11.1%, 3 of 27) had the weak positive results when pneumonia (PCP) was not suspected before COVID-19 pandemic. Two patients had the chronic lung diseases of chronic emphysema and idiopathic pulmonary fibrosis, and one patient did not have abnormal lung parenchymal findings (hospitalized for liver transplantation).

→ Even though we did not include these inpatients in the time-series analyses, we addressed the unnecessary tests in the “Discussion” section in view of the epidemiological importance of potential air circulation of P.jirovecii in inpatients without Pneumocystis pneumonia.

  • [ Discussion: Line -  ] - Even though we did not include the 27 of clinically unnecessary PCR tests in inpatients without the suspicion of PCP in the time-series models, three patients among 22 and 5 tests in the pre- and post-COVID periods had the weak positive results before COVID-19 pandemic. Two patients had the chronic emphysema and idiopathic pulmonary fibrosis, and one patient was hospitalized for SOT without respiratory symptoms or abnormal lung parenchyma.

(3) [2.3]. P. jirovecii PCR: The qualitative real-time PCR tests for P. jirovecii

Q: If the PCR format is real-time PCR, it would mean that the result can be expressed in quantitative cycle, which is much more informative for the circulation of the fungus than a yes or no result. Authors should attempt to quantify their PCR results to analyze transmission. Indeed, even a low fungal load can participate in the transmission (see for instance doi: 10.3389/fmicb.2017.00700).

→ We also think that it’s exactly right point. We checked the cut-off cycle threshold (Ct) values, determining positive, weak positive, and negative, of our qualitative PCP PCR tests.

  : All real-time PCR methods used in our study (40 cycles) determined the positive, weak positive, and negative results as Ct values of <35, 35-37, and >37, respectively.

→ First of all, we addressed the below sentences to clarify Ct values and positive/weak positive results.

  • [ Materials and Methods: 2.4. P. jirovecii PCR, Line:] - All real-time PCR tests for 40 cycles determined the positive, weak positive, and negative results as the cycle threshold (Ct) values of <35, 35-37, and >37, respectively. We defined the positive and weak positive result as the final positive test.

→ And, we re-categorized the results of PCP PCR as positive, weak positive, and negative.

  • Total patients (N = 20,733): total positive – 2,948 (2%) [positive – 2,645 (12.8%), weak positive – 303 (1.4%)], negative – 17,785 (85.8%)
  • Inpatients (N = 20,073): total positive – 2,922 (6%) [positive – 2,624 (13.1%), weak positive – 298 (1.5%)], negative – 17,151 (85.4%)
  • Outpatients (N = 660): total positive – 26 (9%) [positive - 21 (3.2%), weak positive - 5 (0.7%)], negative - 634 (96.1%) (We decided that addition of outpatient’s data does not make much sense in the revised manuscript, because the new very small data (26 and 660 cases during 78 months) did not actually affect the final result and our conclusion in time-serises models based on monthly observed data.

→ Frist of all, we added the data for positive and weak-positive as well as mean Ct values in the Table 2.

→ We described the data of quantitative cycles in the Result section.

  • [ Materials and Methods: 3.1. Clinical information of total PCP-suspected and -confirmed inpatients, Line: ] - The means of quantitative Ct cycles in PCP-confirmed patients were similar between pre- and post-COVID-19 periods (30.9 ± 9.4 vs. 29.1 ± 10.5, P = 0.503). The percentages of weak positive with lower fungal load were only 8.1% and 7.6% in the pre- and post-COVID-19, respectively (Table 2)

→ And, we newly mentioned this point in the Discussion section

  • [ Discussion, Line: ] - Our study should be interpreted in consideration of the following points as probable biases: (1) difficult diagnostic process for P. jirovecii even in patients with symptomatic lung infiltration, (2) not control for the patients who had imaging compatible for PCP and for whom no P. jirovecii test was requested, (3) not surveillance tests in asymptomatic individuals, (4) not all patients are hospitalized, even though our data had very low rate of PCP-confirmed (N=26) and –suspected (N=660) outpatients in the total duration of 78 months, and (5) lack of sophisticated analysis using quantitative PCR values to exam the transmission risk of P. jirovecii with low concentration

  1. Discussion

(1) Lines 310-324: The various forms of P. jirovecii during the life cycle in humans are approximately

…of airborne HCW-to-patient (and vice versa) or person-to-person transmission of P. jirovecii in intra-hospital or community settings.

Q: This long paragraph should be in the introduction, as a rationale for the study. This paragraph does not discuss the results.

→ According to the comment, we moved these sentences into the introduction.

  • [ Introduction - Line: ] - The various forms of P. jirovecii during the life cycle in humans are approximately 1-8 µm in diameter (1-5 µm in trophozite, 4-7 µm in precyst, and 5-8 µm in cyst) [42]. The Korean Central Quarantine Countermeasure Headquarters has recommended four kinds of face masks for everyone-mandatory-mask-wearing policy during the COVID-19 pandemic, which have been approved by the Korean Ministry of Food and Drug Safety (KFDA), including the surgical mask (≥99.9% filtration for 3-µm particles), KF (Korea Filter)-AD (Anti Droplet) (effect in preventing droplet transmission), KF-80 (≥ 80% for 0.6 µm), and KF-94 (≥94% for 0.4 µm, equivalent to the American N95 or European FFP2) (KFDA site: https://www.mfds.go.kr/eng/brd/m_65/view.do?seq=11) [43-45]. These masks can filter large droplets and small aerosols. The surgical mask could be effective in preventing the transmission of SARS-CoV-2 and CA-RVs, including the influenza virus [44,46-48]. The mask adherence rate in South Korea would be fairly high because of the social atmosphere and infringement fines by the local government. Therefore, these basic backgrounds could sufficiently support the effect of mandatory facial mask wearing on the prevention of airborne HCW-to-patient (and vice versa) or person-to-person transmission of P. jirovecii in intra-hospital or community settings.

(2) Line 330- 349

Q: This long paragraph actually discusses the likelihood that "colonized" patients may have played a role. As it is neither explored nor controlled in the present study, the authors should present this paragraph as a limitation of their study. The last part of the paragraph on serology should be in the introduction.

→ According to the comment, we moved the sentence describing the seroepidemiologic studies into the introduction.

  • [ Introduction - Line: ]: Several seroepidemiologic surveys revealed that the majority (70–80%) of healthy children were seropositive for P. jirovecii [36-38].

→ And, we presented the limitation of previous studies.

  • [ Discussion – Line: ]: However, these studies were neither explored nor well controlled. 

(3) Line 350: This first attempt to analyze the potential influence of stricter NPIs during the COVID-19 pandemic on PCP rate has some strengths: (1) complete exclusion of asymptomatic P. jirovecii colonization in PCR test,

Q: As said above, I think it is rather a weakness than a strength when the issue is airborne transmission.

→ We totally agree with this comment. Therefore, we deleted this sentence from the study’s strength part, and added this as our limitation.

  • [ Discussion – “paragraph presenting study strength” Line: ]: This first attempt to analyze the potential influence of stricter NPIs during the COVID-19 pandemic on PCP rate has some strengths: (1) meticulous forecast and interpretation using different time-series prediction models reflecting time variation, particularly, the Bayesian model inferring the causal impact of strict NPIs and/or COVID-19 pandemic on PCP rates, and (2) exclusion of possible biases by change of total number of P. jirovecii tests, especially the underestimation of PCP rates by the increased numbers of tests or suspicious cases.

  • [ Discussion – “paragraph presenting study limitation”Line:  ]: Second, this study could not perform the active surveillance to evaluate the P. jirovecii colonization in inpatients without suspicion of PCP or hospital visitors or HCWs.

(4) Line 358: However, our study has some limitations. First, a major hurdle was the convoluted selection of the exposed or target population to obtain an accurate incidence rate of PCP.

Q: The “convoluted” aspect of the selection should be discussed earlier since it is a major limitation for the study.

→ Along with the reviewer’s previous detailed description for this issue (not hospitalized issue, PCP tests in out-patients, and not control the patients with radiologic findings compatible PCP, and for whom no P. jirovecii test was performed), we did our best to describe this limitation in the another paragraph of the revised discussion section.

  • [4. Discussion - Line: ]: The sentinel surveillance systems are being implemented for seasonal influenza or various communicable infectious diseases in several countries [74,75]. The pathogens with highly contagious and/or public risk should be totally monitored with mandatory reporting [76]. However, the surveillance systems also have the limitation of underreporting or a lapse of surveillance [77]. The exact incidence and prevalence of PCP are difficult to determine, because the large surveillance system for PCP is not available worldwide regardless of the clinical burden and severity. These factors may make it hard to select a target group for our investigation. Our study should be interpreted in consideration of the following points as probable biases: (1) difficult diagnostic process for jirovecii even in patients with symptomatic lung infiltration, (2) not control for the patients who had imaging compatible for PCP and for whom no P. jirovecii test was requested, (3) not surveillance tests in asymptomatic individuals, (4) not all patients are hospitalized, even though our data had very low rate of PCP-confirmed (N=26) and –suspected (N=660) outpatients in the total duration of 78 months, and (5) lack of sophisticated analysis using quantitative PCR values to exam the transmission risk of P. jirovecii with low concentration.

  • [New references]:
  1. 74. Guerrisi, C.; Turbelin, C.; Souty, C.; Poletto, C.; Blanchon, T.; Hanslik, T.; Bonmarin, I.; Levy-Bruhl, D.; Colizza, V. The potential value of crowdsourced surveillance systems in supplementing sentinel influenza networks: The case of france. Euro Surveill 2018, 23.
  2. 75. Mohammed, H.; Hughes, G.; Fenton, K.A. Surveillance systems for sexually transmitted infections: A global review. Curr Opin Infect Dis 2016, 29, 64-69.
  3. 76. Babu Rajendran, N.; Mutters, N.T.; Marasca, G.; Conti, M.; Sifakis, F.; Vuong, C.; Voss, A.; Baño, J.R.; Tacconelli, E. Mandatory surveillance and outbreaks reporting of the who priority pathogens for research & discovery of new antibiotics in european countries. Clin Microbiol Infect 2020, 26, 943.e941-943.e946.
  4. 77. Hsieh, Y.H.; Kuo, M.J.; Hsieh, T.C.; Lee, H.C. Underreporting and underestimation of gonorrhea cases in the taiwan national gonorrhea notifiable disease system in the tainan region: Evaluation by a pilot physician-based sentinel surveillance on neisseria gonorrhoeae infection. Int J Infect Dis 2009, 13, e413-419.

  1. Conclusion

Q: To be changed.

→ We fully consent to this point. This comment is also mentioned by the Reviewer #1 (comment #6 for conclusion)

(‘Taken together our time-series analyses suggest that P. jirovecii may not be mainly acquired by airborne person-to-person transmission’. I think that based exclusively on the results of this study is unwise to assume this affirmation.)

→ Because we total agree with this issue, we deleted the sentence of “Taken together, our time-series analyses suggest that P. jirovecii may not be mainly acquired by airborne person-to-person transmission” in the revised manuscript. And, we inserted the below sentence in the “Conclusion section”

  • Further large studies are needed to uncover the detailed epidemiologic evidences and pathogenic mechanisms of PCP development.
